# Fisher-Rao Sensitivity for Out-of-Distribution Detection in Deep Neural Networks

**Anthony Nguyen,**[*†] **Sylvie Le Hégarat-Mascle, Emanuel Aldea, Renaud Lustrat**[†]
SATIE Laboratory UMR 8029 Paris-Saclay University CNRS, Gif-sur-Yvette, France
{anthony.nguyen,sylvie.le-hegarat,emanuel.aldea}@universite-paris-saclay.fr,
renaud.lustrat@thalesgroup.com

**Antonin Bertrand,**[*‡] **Franck Florin,**[‡] **Mohammed Nabil El-Korso**
L2S Laboratory UMR 8506 Paris Saclay University CNRS, Centrale Supelec, Gif-Sur-Yvette, France
{antonin.bertrand, mohammed.nabil.el-korso}@centralesupelec.fr,
franck.florin@fr.thalesgroup.com

## Abstract

Deep neural networks often remain overconfident on Out-of-Distribution (OoD) inputs. We revisit this problem through Riemannian information geometry. We model the network's predictions as a statistical manifold and find that OoD inputs exhibit higher local Fisher-Rao sensitivity. By quantifying this sensitivity with the trace of the Fisher Information Matrix (FIM), we derive a unifying geometric connection between two common OoD signals: feature magnitude and output uncertainty. Analyzing the limitations of this multiplicative form, we extend our analysis using a product manifold construction. This provides a theoretical framework for the robust additive scores used in state-of-the-art (SOTA) detectors and motivates our final, competitive method.

## 1 Introduction

Deep neural networks have demonstrated remarkable success on a wide range of tasks. However, their reliability often does not extend beyond the training distribution. A major challenge arises when these models are deployed in real-world settings, where they inevitably encounter inputs outside the training distribution. These OoD samples can cause the network to produce predictions that are not only erroneous but also overconfident. This fundamental challenge of distinguishing ID data from OoD inputs is a critical barrier to building truly safe and trustworthy AI systems (Amodei et al., 2016).

A common baseline for OoD detection is to use the model's predictive uncertainty. Hendrycks & Gimpel (2016) showed that, on average, OoD inputs result in lower softmax confidence when fed into a classifier. At the same time, SOTA classifiers can maintain high accuracy under distribution shift. For example, Ovadia et al. (2019) found that as the base model's accuracy increases, OoD accuracy also improves.

To approach this problem from a new perspective, we turn to Information Geometry. A parametrized family of distributions forms a statistical manifold endowed with the Fisher–Rao metric, which is the natural Riemannian metric induced by the model (Gallot et al., 1990; Amari & Nagaoka, 2000). This metric defines the local quadratic form and thus the sensitivity of $p(\cdot, \theta)$ to parameter perturbations. In this work we measure sensitivity via the Fisher metric: we summarize sensitivity with the per–input FIM trace. We provide further discussion in Appendix F about the link between sensitivity and curvature to strengthen our motivation.

---

[*]Co-first authors with equal contribution and importance

[†]Thales Land and Air Systems, BU ARC, Limours, France

[‡]Thales Research & Technology, Palaiseau, France

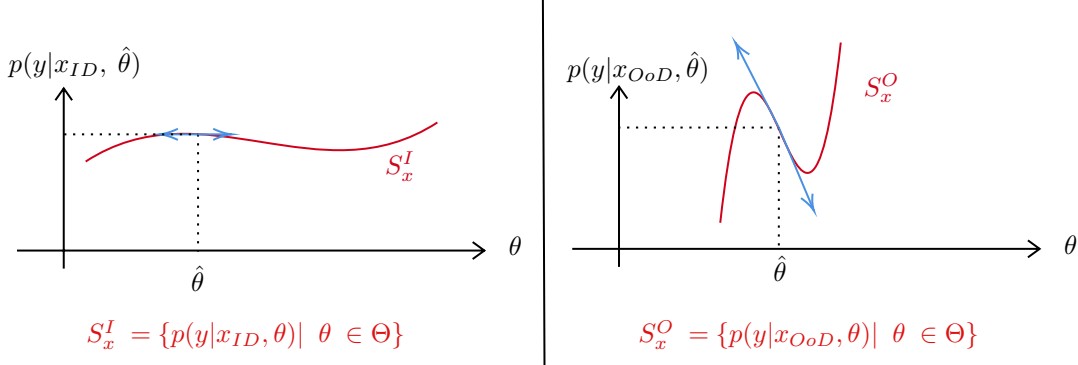

Figure 1: **Fisher–Rao sensitivity distinguishes ID from OoD.** For a fixed input $x$, varying the parameters $\theta$ traces the statistical manifold $S_x = \{p(y|x,\theta) \mid \theta \in \Theta\}$. At the trained parameters $\hat{\theta}$ (dotted line), the manifold $S_x^I$ shows lower local Fisher–Rao sensitivity (left) while $S_x^O$ shows higher local Fisher–Rao sensitivity (right). This higher sensitivity corresponds to a larger size of the elements of the tangent space (blue). See Appendix F for intuitive and technical details on sensitivity vs. curvature.

We use this geometric viewpoint to mathematically characterize the difference between ID and OoD data. We hypothesize and empirically observe that Fisher local sensitivity around a trained model is higher for OoD inputs. Intuitively, a large trace indicates that small parameter perturbations strongly affect the output, while a small trace reflects local stability.

By restricting our analysis to the weights of the final layer, we first establish an analytical expression for the trace, which provides a direct correspondence between the theoretical construct of Fisher-Rao sensitivity and the signals of feature magnitude and output uncertainty. We then generalize this expression by restricting the trace to more discriminative parameter subspaces constructed via a tensor factorization. Finally, motivated by the limitations of a multiplicative form, we propose a more robust additive score derived from a product manifold construction.

In summary, our contributions are:

- **Connecting Geometric Theory to OoD Signals.** Our main result (Theorem 5.1) reveals that, when restricted to the final layer's weights, the intrinsic sensitivity of a classifier neural network decomposes into the product of feature magnitude and output uncertainty, establishing a direct theoretical link between these common signals and Information Geometry.

  **Additive Detectors**: We expose the limitations of the multiplicative form and propose a product manifold construction, leading to an additive score. In doing so, we provide a geometric justification for the additive composition found in SOTA methods (e.g., (Wang et al., 2022; Igoe et al., 2022)), leading to our final competitive detector.

- **Competitive performance.** Our additive detector require only a single forward pass, with no OoD data exposure and with no training or inference-time modifications, *i.e.*, the base model is used as is. It achieves competitive results against related post-hoc baselines.

## 2 RELATED WORK

OoD detection methods are often divided into two categories: those that modify the training process and those that are *post-hoc*.

One line of work integrates OoD awareness directly into training. This is done through techniques such as outlier exposure during training (Hendrycks et al., 2019; Zhang et al., 2023; Gao & Li, 2025) or data augmentation (Zhang et al., 2017; Hendrycks et al., 2022). Other works consider architectural choices that prevent overconfidence (Hein et al., 2019), or methods that shape the feature space, such as activation shaping (Sun et al., 2021; Djurisic et al., 2023) or controlling the model's Lipschitz constant (Liu et al., 2020a; Mukhoti et al., 2023). Others leverage geometrical properties (Haas et al., 2023; Regmi et al., 2024; Nguyen et al., 2024; 2025) of the feature space. While very effective, these methods require specific training protocols or test-time alterations.

Post-hoc methods are widely used due to their flexibility. These can be broadly grouped into two sub-categories. The first involves **inference-time modifications**, where the network's forward pass is altered.

A popular approach is activation shaping (e.g., ReAct (Sun et al., 2021), DICE (Sun & Li, 2022) or ASH (Djurisic et al., 2023)), which clips or sparsifies feature activations before they are passed to the final layer. While highly effective, these techniques alter the network's computations.

In contrast, our proposed method belongs to the second sub-category of **purely analytical methods**, providing an OoD score without any modification during inference. Within this analytical category, many methods analyze the geometry of the model's latent feature space, typically by measuring distances to class prototypes. Examples include Mahalanobis (Lee et al., 2018), Deep k-NN (Sun et al., 2022), and ViM (Wang et al., 2022). Other distance approaches adopt a deeper geometric perspective. This includes IGEOOD (Gomes et al., 2022), which uses the Fisher-Rao geodesic distance on the statistical manifold whose geometry is fundamentally defined by the gradients of the model's log-likelihood. The gradient is leveraged as a geometric tool by other methods in different ways. For instance, ODIN (Liang et al., 2017) perturbs the input using the gradient of the classification loss to amplify the difference in softmax outputs between ID and OoD samples. GradNorm (Huang et al., 2021) uses the gradient norm with respect to a uniform distribution as an OoD score. However, the role of gradients in these methods has been questioned. In their empirical analysis of Grad-Norm, Igoe et al. (2022) argue that the method's success comes from a simple combination of two signals: the magnitude of the feature encoding and the statistics of the output distribution. More recently, GradOrth (Behpour et al., 2023) extends this by projecting gradients onto a learned subspace. Similarly, Kwon et al. (2020) use backpropagated gradients as Fisher Kernels to characterize the local geometry of the autoencoder's manifold, requiring training-time regularization to constrain the tangent space. To further contextualize our method, we compare in Appendix C our first baseline score, the **Standard FIM Trace**, against GradNorm (Huang et al., 2021) and IGEOOD.

## 3 INFORMATION-GEOMETRY PRELIMINARIES

### 3.1 STATISTICAL MODEL AND RIEMANNIAN GEOMETRY

Information geometry provides a rigorous framework for analyzing the structure of statistical models by modeling them as a Riemannian manifold (Amari, 2016). We briefly recall the key concepts before specializing them to predictive models $p(y|x, \theta)$ used in deep learning.

**Definition 3.1** (Statistical manifold). *Let $(\mathcal{X}, \mathcal{A}, \mu)$ be a $\sigma$-finite measure space and $\Theta \subset \mathbb{R}^m$ open. Consider an identifiable, $\mu$-dominated model $\{P_\theta\}_{\theta \in \Theta}$ with densities $p(\cdot, \theta)$ that are $C^\infty$ in $\theta \in \Theta$. Then*

$$S = \{p(\cdot \, ; \theta) \mid \theta \in \Theta\}$$

*is an $m$-dimensional $C^\infty$ manifold, via the parametrization $\varphi : \Theta \to S$, $\varphi(\theta) = p(\cdot \, ; \theta)$.*

By identifiability, the parametrization $\varphi$ is one-to-one, so $\theta$ serves as a global coordinate chart on $S$. To quantify the effect of parameter perturbations, the manifold is endowed with a **Riemannian metric**, which defines an inner product on each tangent space, allowing for the measurement of distances and angles. A fundamental result in information geometry is that the **Fisher Information Matrix** serves as a natural Riemannian metric for a statistical manifold.

**Definition 3.2** (Fisher-Rao Metric). *The **Fisher Information Matrix** (Rao, 1992) is defined by its components which are given by the following expectation:*

$$F_{ij}(\theta) = \mathbb{E}_{u \sim p(\cdot, \theta)} \left[ \frac{\partial \log p(u, \theta)}{\partial \theta_i} \frac{\partial \log p(u, \theta)}{\partial \theta_j} \right]. \tag{1}$$

This metric is invariant to smooth reparameterization of $\theta$ and provides an intrinsic measure of statistical distinguishability through the quadratic form $d\theta^\top F(\theta)\, d\theta$ where $(\cdot)^\top$ denotes the transpose operator.

## 3.2 LOCAL SENSITIVITY VIA THE TRACE OF THE FISHER INFORMATION MATRIX

On the Riemannian statistical manifold $(S, F)$, infinitesimal parameter changes $d\theta$ yield the squared (statistical) distance $ds^2 = d\theta^\top F(\theta)\, d\theta$, which measures how sensitive $p(\cdot, \theta)$ is to infinitesimal changes of the parameters.

**Proposition 3.3** (Trace of the FIM). *The trace of the FIM $F(\theta)$, defined as the sum of its diagonal elements, is given by the expected squared L2 norm of the score vector:*

$$\mathrm{Tr}\left(F(\theta)\right) \;=\; \mathbb{E}_{u \sim p(\cdot, \theta)}\left[\,\|\nabla_\theta \log p(u, \theta)\|_2^2\,\right]. \tag{2}$$

## 3.3 FROM FISHER–RAO GEOMETRY TO DEEP NEURAL NETWORKS

The theoretical framework of information geometry can be directly applied to deep neural networks. A neural network with a fixed architecture is specified by its parameter vector $\theta \in \mathbb{R}^m$. Each parameter vector defines a conditional probability distribution $p(y|x, \theta)$. For any input $x$, the set of all such distributions forms a statistical manifold $S_x$, parametrized by $\theta \in \mathbb{R}^m$:

$$S_x = \{p(y|x, \theta) \mid \theta \in \mathbb{R}^m\}.$$

Thus, a specific model is a point on the manifold $S_x$. The tangent space at that point, $T_{p_\theta} S_x$, represents all possible infinitesimal changes $d\theta$. The Fisher-Rao metric, which is input-wise, provides a natural way to measure, for each input, the geometric and statistical impact of these weight perturbations.

## 4 DEEP LEARNING/OoD BACKGROUND AND NOTATIONS

### 4.1 DATA, DISTRIBUTIONS, AND ID/OoD MODEL

Let the random variables $X \in \mathcal{X} \subset \mathbb{R}^d$ and $Y \in \mathcal{Y} = \{1, \ldots, K\}$ denote the input and label, respectively. We assume an unknown joint distribution $p_{XY}$ on $\mathcal{X} \times \mathcal{Y}$ and an ID training dataset $\mathcal{D}_{\mathrm{train}} = \{(x_i, y_i)\}_{i=1}^N \sim p_{XY}$. The ID input marginal is $p_X$. To model open-world inputs, we introduce a binary random variable $Z \in \{0, 1\}$ indicating whether a test input is ID ($Z{=}0$) or OoD ($Z{=}1$). The test-time input distribution is a mixture with $p_{X|Z}(x \mid 0) = p_X(x)$ and $p_{X|Z}(x \mid 1) = q_X(x)$, where $q_X$ is an arbitrary (unknown) OoD marginal. We do not assume any parametric form for $p_X$ or $q_X$. We denote the sets of ID and OoD inputs by $\mathcal{X}_{\mathrm{ID}}$ and $\mathcal{X}_{\mathrm{OoD}}$, respectively.

### 4.2 NEURAL NETWORK PREDICTOR

A classifier with parameters $\theta \in \mathbb{R}^m$ defines the distribution $p(y|x, \theta) = \mathrm{softmax}\left(g_\theta(f_\theta(x))\right)$ where we decompose the network into a feature extractor $f_\theta : \mathcal{X} \to \mathbb{R}^D$ and a classification head $g_\theta : \mathbb{R}^D \to \mathbb{R}^K$ producing logits in $\mathbb{R}^K$. We denote the predictor $h_\theta = g_\theta \circ f_\theta$. After training on $\mathcal{D}_{\mathrm{train}}$ we obtain $\hat{\theta}$ and the predicted label $\hat{y}(x) = \arg\max_{y \in \mathcal{Y}} p(y|x, \hat{\theta})$.

**Notation at a fixed trained model.** We fix a pre-trained classifier with parameters $\hat{\theta} \in \mathbb{R}^m$ and conditional distribution $p(y|x, \hat{\theta})$ over $K$ classes. We adopt the standard setting where the classification head is a single linear layer applied to the penultimate representation: let $f_{\hat{\theta}}(x) \in \mathbb{R}^D$ denote the penultimate feature, and let the predictor be $h_{\hat{\theta}}(x) = W^\top f_{\hat{\theta}}(x) + b$ where $W \in \mathbb{R}^{D \times K}$ and $b \in \mathbb{R}^K$ are the parameters of this final layer and constitute a subset of the full parameter vector $\hat{\theta}$. Then $p(y|x, \hat{\theta}) = \mathrm{softmax}\left(h_{\hat{\theta}}(x)\right)$. For convenience we write $p_{\hat{\theta},k}(x) = p(y{=}k|x, \hat{\theta})$ and $\|p_{\hat{\theta}}(x)\|_2^2 = \sum_{k=1}^K p_{\hat{\theta},k}(x)^2$.

**Information Geometry Notation.** According to Section 3.3, for any $x \in \mathcal{X}$, the input-specific Fisher Information Matrix is defined as the expectation over the conditional distribution:

$$F_x(\theta) = \mathbb{E}_{y \sim p(\cdot|x,\theta)} \left[ \nabla_\theta \log p(y|x,\theta) \nabla_\theta \log p(y|x,\theta)^\top \right]. \tag{3}$$

Its trace is given by:

$$\mathrm{Tr}\left(F_x(\theta)\right) = \mathbb{E}_{y \sim p(\cdot|x,\theta)} \left[ \|\nabla_\theta \log p(y|x,\theta)\|_2^2 \right]. \tag{4}$$

To analyze the model's sensitivity specifically to its last-layer weights $W$, we focus on the corresponding block of the FIM. This block, denoted $F_{W,x}(\theta)$, is defined as:

$$F_{W,x}(\theta) = \mathbb{E}_{y \sim p(\cdot|x,\theta)} \left[ \nabla_W \log p(y|x,\theta)(\nabla_W \log p(y|x,\theta))^\top \right]. \tag{5}$$

# 5 TRACE OF FIM FOR OoD CHARACTERIZATION

## 5.1 MOTIVATION

We approach OoD detection by examining the stability induced during optimization. We posit that the training process may encourage the learned parameters $\hat{\theta}$ to be relatively stable for the ID distribution, phenomenon related to stochastic optimizers where small parameter perturbations have minimal impact on predictions for ID data (Maddox et al., 2019). In other words, small perturbations of $\hat{\theta}$ are unlikely to significantly affect predictions for ID inputs. However, since this stability is only learned from ID data, it may not hold for OoD inputs. Consequently, the local geometry of the manifold $S_{x_{\mathrm{OoD}}}$ can exhibit higher sensitivity for $x_{\mathrm{OoD}}$. We quantify this sensitivity with the per-input FIM trace $\mathrm{Tr}(F_x(\hat{\theta}))$ introduced above (Figure 1).

In practice, computing the trace of the full FIM over all parameters $\hat{\theta}$ is intractable for deep neural networks. Therefore, we focus on the parameters of the final layer, as they directly control how the model translates high-level features into the final prediction (Pearce et al., 2021).

## 5.2 SPECIALIZATION TO PENULTIMATE-LAYER SUBSPACES

Let $\mathcal{U} \subset \mathbb{R}^m$ be an $r$-dimensional linear subspace of the parameter space. We represent this subspace with a matrix $P \in \mathbb{R}^{m \times r}$ whose columns form an orthonormal basis for $\mathcal{U}$. Any parameter perturbation $d\theta$ that lies within this subspace can be expressed as $d\theta = Pd\alpha$, where $d\alpha \in \mathbb{R}^r$ is the coordinate vector in basis $P$. Substituting into the general Fisher-Rao metric yields, for any $x \in \mathcal{X}$, the metric induced on the subspace $\mathcal{U}$:

$$ds^2 := (Pd\alpha)^\top F_x(\hat{\theta})(Pd\alpha) = d\alpha^\top \left( P^\top F_x(\hat{\theta})P \right) d\alpha. \tag{6}$$

The matrix $P^\top F_x(\hat{\theta})P$ is the restriction of the FIM to $\mathcal{U}$ and for any $x \in \mathcal{X}$, we have $F_{W,x}(\hat{\theta}) = \mathrm{Tr}\left(P^\top F_x(\hat{\theta})P\right)$.

**Theorem 5.1** (Closed-Form FIM Trace for the Final Layer). *Take $\mathcal{U}$ to be the coordinate subspace spanned by the vectorized last-layer weights $W$. We have the following closed-form expression:*

$$\mathrm{Tr}\left(F_{W,x}(\hat{\theta})\right) = \|f_{\hat{\theta}}(x)\|_2^2 \left( 1 - \|p_{\hat{\theta}}(x)\|_2^2 \right). \tag{7}$$

*Proof sketch.* For the linear output layer, the gradient factorizes as the product $\nabla_W \log p = (e_y - p_{\hat{\theta}})f_{\hat{\theta}}^\top$. Substituting this into the trace, the squared feature norm $\|f_{\hat{\theta}}\|_2^2$ factors out. The remaining term is the expectation of the squared prediction error $\mathbb{E}_y[\|e_y - p_{\hat{\theta}}\|_2^2]$, which corresponds to the multinomial variance $1 - \|p_{\hat{\theta}}\|_2^2$. For full proof, see Appendix E. $\square$

We will refer to equation 7 as the Standard FIM Trace.

**Connecting Fisher-Rao Sensitivity to common OoD Signals.** This theorem is foundational as it reveals that the geometric notion of Fisher-Rao sensitivity, when applied to a neural network's final layer, manifests as a product of the feature vector's magnitude and the output distribution's uncertainty. This provides a bridge between geometry and the design of several existing heuristic-based methods (e.g., GradNorm, Energy-based scores) which combine similar signals.

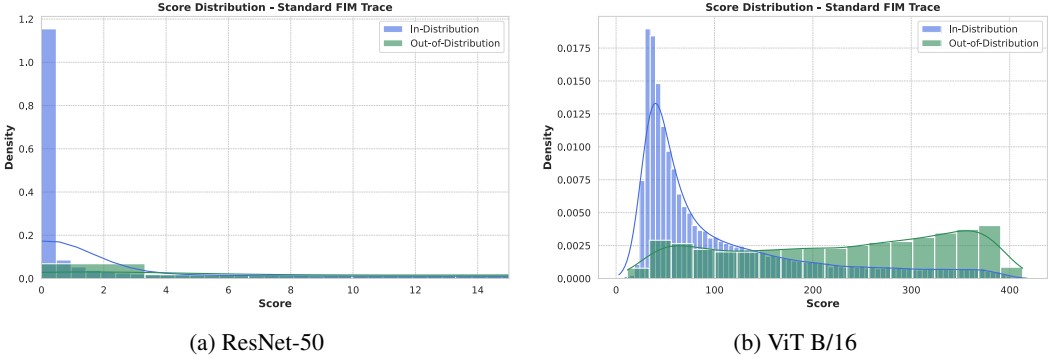

(a) ResNet-50                           (b) ViT B/16

Figure 2: **Empirical validation of the Fisher Rao Sensitivity.** Score distributions for the Standard FIM Trace (equation 7) evaluated on a ResNet-50 (left) and ViT (right), with ImageNet as ID (blue) and Places365 as OoD (green). The plot shows a visible shift in the OoD distribution towards higher values, empirically confirming that OoD inputs tend to exhibit greater local Fisher-Rao sensitivity. However, the imperfect overlap between the distributions indicates the limitations of this baseline score, motivating the more advanced subspace methods developed in Section 5.3.

### 5.2.1 DISCRIMINATIVE SUBSPACE VIA TENSOR FACTORIZATION

The Standard FIM Trace considers the entire parameter space of the final layer, treating all directions of perturbation equally. However, as suggested by the overlap in Figure 2, not all directions are equally informative for separating ID from OoD data. For instance, it can be uninformative when both the embedding norm and the softmax flatness fail to separate ID from OoD. Our key insight is to use more discriminative lower-dimensional subspaces where the separation is enhanced.

To construct such a subspace, we leverage the natural structure of the last layer's gradients. For any input $x$, the gradient of the log-probability with respect to the last-layer weights $W$ factorizes into a class-dependent term and a feature-dependent term. Specifically, its vectorized form is a tensor product (Bourbaki, 2006):

$$\text{vec}(\nabla_W \log p(y|x,\hat{\theta})) = \underbrace{(e_y - p_{\hat{\theta}}(x))}_{\text{class part}} \otimes \underbrace{f_{\hat{\theta}}(x)}_{\text{feature part}}, \tag{8}$$

where $e_y \in \mathbb{R}^K$ is the $y$-th canonical basis vector (See Appendix G). This decomposition of the gradient, which forms the basis of the FIM, allows us to decompose the parameter space. We can thus select informative directions for the class and feature spaces.

This motivates defining a subspace using a tensor product basis $P = Q \otimes R$, where the columns of $Q \in \mathbb{R}^{K \times r_c}$ span a class subspace and the columns of $R \in \mathbb{R}^{D \times r_f}$ span a feature subspace. The methods for constructing these projection matrices $Q$ and $R$ using PCA and LDA are detailed at the end of Section 5.3. This approach leads to the following result for the FIM trace restricted to this specialized subspace.

**Proposition 5.2** (Tensor FIM Trace). *Let $P = Q \otimes R$ and $S_{p_{\hat{\theta}}(x)} = \text{diag}(p_{\hat{\theta}}(x)) - p_{\hat{\theta}}(x)p_{\hat{\theta}}(x)\top$. For any $x \in \mathcal{X}$, the restricted FIM trace for the last layer is:*

$$\text{Tr}(P^\top F_{W,x}(\hat{\theta})P) = \underbrace{\text{Tr}(Q^\top S_{p_{\hat{\theta}}(x)} Q)}_{U_{\hat{\theta}}(x)} \cdot \underbrace{\|R^\top f_{\hat{\theta}}(x)\|_2^2}_{M_{\hat{\theta}}(x)}. \tag{9}$$

See proof in Appendix E.

### 5.3 FROM A SINGLE MANIFOLD TO A PRODUCT MANIFOLD

The Standard FIM Trace in equation 9 reveals that local sensitivity manifests as a product of predictive uncertainty $U_{\hat{\theta}}(x)$ and feature magnitude $M_{\hat{\theta}}(x)$. However, this multiplicative coupling can be suboptimal, as one weak signal can suppress the other (see Appendix Dand Igoe et al. (2022)).This

limitation motivates us to decouple these signals, treating them as structurally independent sources of geometric sensitivity. To explicitly model the uncertainty and feature space components as orthogonal axes of variation, **the canonical geometric structure to model the parameter space is a product manifold**.

### 5.3.1 Geometrical Construction and Properties of the Additive Score

To construct this product manifold, we first need to refine the geometric description of the feature space. The magnitude term $M_{\hat{\theta}}(x)$ only captures the feature's alignment with the learned ID subspace, which is an incomplete geometric description. A more complete picture requires considering the component of the feature that lies *orthogonal* to this subspace. The Pythagorean theorem provides a natural way for this, decomposing the feature's total energy into two orthogonal parts for any orthonormal matrix $R$:

$$\|f_{\hat{\theta}}(x)\|_2^2 = \underbrace{\|R^\top f_{\hat{\theta}}(x)\|_2^2}_{\text{Proj. Magnitude}} + \underbrace{\|f_{\hat{\theta}}(x) - RR^\top f_{\hat{\theta}}(x)\|_2^2}_{\text{Residual Energy}} \tag{10}$$

This decomposition naturally reveals the residual energy, the component of the feature unexplained by the learned subspace. The orthogonality of the projected magnitude and residual energies allows us to treat them as independent geometric signals.

Building on this orthogonality assumption, we model the total sensitivity on a product manifold $M := \mathcal{U} \times \mathcal{U} \times \mathcal{U}$. We define generalized and specialized metrics $(F', F'', F''')$ on these three copies of the parameter subspace $\mathcal{U}$ via (pseudo)-conformal transformations (see Appendix F.3) based on the projected FIM. We suppose that $\forall x \in \mathcal{X}$, $M_{\hat{\theta}}(x)$, $U_{\hat{\theta}}(x)$ and $\frac{(\|f_{\hat{\theta}}(x)\|_2^2 - M_{\hat{\theta}}(x))}{(U_{\hat{\theta}}(x) \cdot M_{\hat{\theta}}(x))}$ are non-zero. Let $(\lambda_M, \lambda_y) \in (\mathbb{R}^*)^2$. We define:

$$F'_{W,x}(\hat{\theta}) := \frac{F_{W,x}(\hat{\theta})}{M_{\hat{\theta}}(x)}, F''_{W,x}(\hat{\theta}) := \frac{F_{W,x}(\hat{\theta})}{\lambda_M^{-1} U_{\hat{\theta}}(x)}, F'''_{W,x}(\hat{\theta}) := \frac{\left(\|f_{\hat{\theta}}(x)\|_2^2 - RR^T f_{\hat{\theta}}(x)\|_2^2\right) F_{W,x}(\hat{\theta})}{\lambda_y^{-1}(U_{\hat{\theta}}(x) \cdot M_{\hat{\theta}}(x))}, \tag{11}$$

such that their traces are precisely our three signals of interest: $\text{Tr}(F'_{W,x}(\hat{\theta})) = U_{\hat{\theta}}(x)$, $\text{Tr}(F''_{W,x}(\hat{\theta})) = \lambda_M M_{\hat{\theta}}(x)$, and $\text{Tr}(F'''_{W,x}(\hat{\theta})) = \lambda_y(\|f_{\hat{\theta}}(x) - RR^T f_{\hat{\theta}}(x)\|_2^2) = \lambda_y y_{\hat{\theta}}(x)$. Formally, we evaluate the local sensitivity at the diagonal injection of the learned parameters into the product space, i.e., at the point $(\hat{\theta}, \hat{\theta}, \hat{\theta}) \in M$. At this point, the tangent space decomposes into a direct sum of the component tangent spaces $T_\Theta M \cong T_{\hat{\theta}}\mathcal{U} \oplus T_{\hat{\theta}}\mathcal{U} \oplus T_{\hat{\theta}}\mathcal{U}$. Consequently, a perturbation vector in $M$ is a tuple of independent perturbations, and the natural metric on $M$ is the block-diagonal tensor representing the direct sum of the component metrics:

$$F^{\oplus}_{W,x}(\hat{\theta}) := \begin{pmatrix} F'_{W,x}(\hat{\theta}) & 0 & 0 \\ 0 & F''_{W,x}(\hat{\theta}) & 0 \\ 0 & 0 & F'''_{W,x}(\hat{\theta}) \end{pmatrix}. \tag{12}$$

Its trace, representing the total sensitivity on $M$, is the sum of the traces of its components:

$$\begin{aligned} \text{Tr}(F^{\oplus}_{W,x}(\hat{\theta})) &= \text{Tr}(F'_{W,x}(\hat{\theta})) + \text{Tr}(F''_{W,x}(\hat{\theta})) + \text{Tr}(F'''_{W,x}(\hat{\theta})) \\ &= \underbrace{U_{\hat{\theta}}(x)}_{\text{Uncertainty}} + \lambda_M \underbrace{M_{\hat{\theta}}(x)}_{\text{Proj. Magnitude}} + \lambda_y \underbrace{(\|f_{\hat{\theta}}(x) - RR^T f_{\hat{\theta}}(x)\|_2^2)}_{\text{Residual } y_{\hat{\theta}}(x)}. \end{aligned} \tag{13}$$

$(\lambda_M, \lambda_y)$ are hyperparameters that balance the scales of these contributions as we do not have access to OOD labels during tuning. Without access to OoD correlations during tuning, we adopt the standard principle of indifference: we tune the hyperparameters assuming the components are independent. These are determined by minimizing:

$$\mathcal{L}(\lambda_y, \lambda_M) = \left(\text{Var}[\lambda_y \cdot y_{\hat{\theta}}(x)] - \text{Var}[U_{\hat{\theta}}(x)]\right)^2 + \left(\text{Var}[\lambda_M \cdot M_{\hat{\theta}}(x)] - \text{Var}[U_{\hat{\theta}}(x)]\right)^2 \tag{14}$$

This objective is a 4th-degree polynomial in $\lambda_y$ and $\lambda_M$ and immediately yields the analytical solution:

$$|\lambda_y| = \sqrt{\text{Var}(U_{\hat{\theta}}(x))/\text{Var}(y_{\hat{\theta}}(x))} \quad \text{and} \quad |\lambda_M| = \sqrt{\text{Var}(U_{\hat{\theta}}(x))/\text{Var}(M_{\hat{\theta}}(x))} \tag{15}$$

and the choice of the signs are based on the signal contribution analysis detailed in Appendix D. In the case where the hyperparameter is strictly negative, the manifold is not Riemannian. However we get a pseudo-Riemannian manifold structure where the local sensitivity is still well-defined. See Appendix F.

**Proposition 5.3** (Variance Balancing under Independence Hypothesis). *If we further assume statistical independence of the uncertainty $U$ and feature-space signals $(M, y)$, minimizing the objective $\mathcal{L}$ guarantees that each component contributes equally to the total variance of the additive score.*

*Proof.* See proof in Appendix E. □

A rigorous geometric measure such as Fisher-Rao sensitivity must capture the intrinsic local curvature of the statistical manifold independently of parameterization. The following proposition shows that our score remains consistent across equivalent models regardless of their specific parameterization.

**Proposition 5.4** (Invariance under Reparameterization). *The additive score, when tuned via equation 14, is invariant under the group of similarity transformations acting on the feature space $f_{\hat{\theta}}$ and weights $W$ that preserve the predictive distribution $p(y|x, \hat{\theta})$:*

1. **Orthogonal Transformation:** $f_{\hat{\theta}} \rightarrow \mathbf{R} f_{\hat{\theta}}$ and $W \rightarrow \mathbf{R}W$, where $\mathbf{R}^{\top}\mathbf{R} = I$ (e.g., rotation).

2. **Classifier Translation:** $W \rightarrow W + \mathbf{C}$, where $\mathbf{C} \in \mathbb{R}^{D \times K}$ has identical columns (shift of logits).

3. **Uniform Scaling:** *The joint rescaling* $f_{\hat{\theta}} \rightarrow \alpha f_{\hat{\theta}}$ and $W \rightarrow \alpha^{-1}W$, with $\alpha \neq 0$.

*Proof.* See proof in Appendix E. □

This invariance holds even when the subspaces $Q$ and $R$ are estimated from the data. For instance, applying a rotation to the input features results in an equivalent rotation of the estimated eigenvectors, leaving the projected norms unchanged.

**Subspace Matrices Construction.** Our method relies on the construction of two discriminative subspaces from a random chosen calibration set sampled from ID data, $\mathcal{D}_{\text{val}}$. The selection of algorithms to construct the projection matrices for the feature space $R$ and the probability space $Q$ is deliberately chosen to align with the distinct geometric properties of each space. For the feature space, we construct the matrix $R$ using Linear Discriminant Analysis (LDA) followed by an orthonormalization process. The basis vectors are derived from the feature representations $\{f_{\hat{\theta}}(x)\}_{x \in \mathcal{D}_{\text{val}}}$ and their corresponding class labels. The choice of LDA is to find a projection that maximizes the inter-class variance while simultaneously minimizing the intra-class variance. Our hypothesis is that feature vectors of ID inputs will exhibit a high projection norm onto this manifold, whereas OoD features, lacking correspondence to the learned class structure, will be characterized by a significant orthogonal component. For the probability space, we construct the matrix $Q$ using Principal Component Analysis (PCA) on the set of softmax distributions $\{p_{\hat{\theta}}(x)\}_{x \in \mathcal{D}_{\text{val}}}$. This subspace models the typical patterns of predictive uncertainty and inter-class confusion exhibited by the model on ID data. It is the natural linear algebraic tool designed to analyze such covariance structures: it diagonalizes the covariance matrix to identify the orthogonal directions of maximum variance: it extracts the principal axes of the Fisher Information Matrix (the directions of highest sensitivity) in a linear basis. Therefore, the power of this score relies on a two-stage process: first, learning a discriminative subspace to create a meaningful geometric decomposition, and second, tuning the weights $\lambda_M$ and $\lambda_y$ to find the optimal balance between the magnitude and residual signals.

## 6 EXPERIMENTAL RESULTS

We conduct experiments on standard OoD detection benchmarks to validate our geometric framework. The evaluation is designed to answer two questions from our theory: 1) Does our final, additive score, motivated by the product manifold construction, achieve competitive performance

against analytic post-hoc baselines? 2) Does this additive formulation provide a significant empirical advantage over the simpler multiplicative score obtained from our initial analysis?

**Datasets and models.** We consider two experimental settings. CIFAR: ResNet-18 (He et al., 2016) trained on CIFAR-10/100 (Krizhevsky et al., 2009), the OoD datasets are given by the complementary CIFAR dataset and SVHN (Netzer et al., 2011). ImageNet-1K: ResNet-50 or ViT-B/16 (Dosovitskiy et al., 2020) trained on ImageNet-1K (Deng et al., 2009), the OoD datasets include Places365 (Zhou et al., 2017), ImageNet-O (Hendrycks et al., 2021), iNaturalist (Van Horn et al., 2018) and SUN (Xiao et al., 2010).

Table 1: OOD Detection Performance (AUROC / TNR@TPR95) on ImageNet benchmark

| ViT-B/16 | | | | |
| --- | --- | --- | --- | --- |
| **Method** | **ImageNet-O** | **iNaturalist** | **Places365** | **SUN** |
| MSP | 85.08/46.89 | 96.64/87.91 | 84.76/43.67 | 86.62/46.14 |
| Energy Score | 91.72/67.28 | 97.85/94.58 | 89.05/54.21 | 90.57/61.73 |
| GradNorm | 82.23/63.92 | 96.11/92.90 | 82.68/52.79 | 85.57/40.21 |
| GradOrth | **93.22**/67.98 | **98.98/98.49** | 89.78/**61.10** | 93.15/**68.68** |
| Mahalanobis Distance | 91.94/**68.51** | 98.76/97.88 | 88.75/57.12 | 90.07/61.17 |
| ODIN | 90.23/67.31 | 98.45/93.45 | 87.27/49.18 | 91.46/61.39 |
| Deep k-NN | 92.21/67.48 | 98.34/98.37 | 88.06/53.42 | 92.56/65.70 |
| ViM | 92.26/71.34 | 98.87/98.15 | **90.83**/58.07 | 93.32/65.36 |
| IGEOOD | 91.66/66.27 | 97.89/97.56 | 87.31/52.12 | 90.32/60.66 |
| Standard FIM Trace (Ours) | 89.03/65.89 | 96.92/91.34 | 86.28/51.66 | 88.56/58.72 |
| Tensor FIM Trace (Ours) | 90.12/62.34 | 97.27/94.18 | 86.92/55.56 | 90.23/63.78 |
| Additive FIM Trace (Ours) | 92.61/68.25 | 98.82/98.41 | 89.51/60.17 | **93.38**/67.98 |
| ResNet-50 | | | | |
| **Method** | **ImageNet-O** | **iNaturalist** | **Places365** | **SUN** |
| MSP | 58.23/15.98 | 88.08/50.41 | 78.11/30.10 | 79.47 30.18 |
| Energy Score | 59.28/15.91 | 89.29/47.31 | 77.47/29.12 | 79.23/29.22 |
| GradNorm | 60.36/18.98 | 67.54/28.35 | 70.96/13.53 | 69.17/04.13 |
| GradOrth | 71.23/**23.56** | 90.12/52.42 | 82.43/39.64 | 85.12/**40.34** |
| Mahalanobis Distance | 69.11/21.43 | 52.01/39.17 | 72.71/29.73 | 54.56/18.23 |
| ODIN | 64.34/16.21 | 89.14/40.65 | 77.56/31.22 | 78.31/24.51 |
| Deep k-NN | **72.34**/22.82 | 86.97/37.81 | 70.51/30.25 | 81.94/24.83 |
| ViM | 68.21/22.34 | 87.19/45.79 | 79.66/36.94 | 76.84/27.35 |
| IGEOOD | 67.43/22.50 | 87.18/47.02 | 82.45/37.47 | 72.23/05.63 |
| Standard FIM Trace (Ours) | 61.11/18.62 | 89.74/49.09 | 78.92/29.16 | 83.23/19.13 |
| Tensor FIM Trace (Ours) | 65.23/21.71 | 89.89/52.23 | 81.46/36.23 | 83.78/31.45 |
| Additive FIM Trace (Ours) | 71.77/22.41 | **90.61/54.69** | **82.93/40.33** | **85.76**/39.95 |

**Baselines.** We compare our methods against a comprehensive set of **purely analytic** post-hoc OoD detection methods, including uncertainty-based methods such as MSP (Hendrycks & Gimpel, 2016), Energy (Liu et al., 2020b), and gradient-based methods such as ODIN (Liang et al., 2017), GradNorm (Huang et al., 2021) and GradOrth (Behpour et al., 2023). We include IGEOOD as a recent related information-geometric (Gomes et al., 2022) baseline and related feature-space distance-based methods: Mahalanobis (Lee et al., 2018), Deep k-NN (Sun et al., 2022) and ViM (Wang et al., 2022) as a SOTA additive baseline score. We emphasize that ViM also combine feature with logits signals through a virtual class construction and we provide a comparison in Appendix C.

**Ablation Study.** We provide a detailed ablation study in Appendix A using a ResNet-50 on ImageNet. In particular, we highlight the superiority of LDA over PCA for creating the feature subspace and also demonstrate that our method is robust to the size and the sampling of the calibration set.

**Implementation Details.** All methods evaluated in this work are post-hoc and do not require retraining the base model nor inference-time modifications. For our proposed methods, the projection matrices Q and R are constructed using a calibration set (50,000 images for ImageNet; 5,000 for CIFAR) sampled from the ID training data. To ensure a fair comparison, the hyperparameters for all baseline methods were tuned on the same validation set.

**Evaluation Metrics.** We report results using standard OoD detection metrics: 1) The Area Under the Receiver Operating Characteristic curve (AUROC) which is our primary metric; 2) The True Negative Rate at $95\%$ True Positive Rate (TNR@TPR95). In both cases, higher values indicate better performance.

**Performance on Benchmarks.** The empirical results serve to validate our geometric framework. As shown in Table 2, our method is competitive and stable on the CIFAR benchmarks. On the more challenging near-OoD settings (CIFAR-10 vs. CIFAR-100 and vice-versa), our proposal achieves competitive results. While some baselines vary substantially across datasets, our additive score remains consistently effective, demonstrating a reliability similar to robust methods like ViM. The Additive FIM Trace provides a clear improvement over the Standard and Tensor FIM Trace, empirically validating the theoretical motivation for moving from a multiplicative to an additive score. As detailed in Table 1, this strong performance extends to the large-scale ImageNet-1K benchmark. Our additive score is competitive against strong post-hoc methods on both ResNet-50 and ViT-B/16 architectures, confirming that the principles derived from our framework are applicable across different models and data.

Table 2: OOD Detection Performance (AUROC/TNR@TPR95) on CIFAR benchmarks

| Method | ID: CIFAR-10 | | ID: CIFAR-100 | |
|---|---|---|---|---|
| | **vs. SVHN** | **vs. C-100** | **vs. C-10** | **vs. SVHN** |
| MSP | 85.48/65.80 | 83.35/36.61 | 70.33/14.57 | 74.93/35.61 |
| Energy Score | 88.61/67.72 | 85.57/32.26 | 70.40/14.77 | 76.71/36.81 |
| GradNorm | 85.72/63.17 | 82.11/27.12 | 70.28/13.99 | 74.18/19.56 |
| GradOrth | 89.11/67.23 | 86.99/54.24 | **74.42**/14.78 | **81.92**/**37.23** |
| Mahalanobis Distance | 89.63/66.22 | 70.53/24.21 | 58.73/04.35 | 80.45/24.12 |
| ODIN | 87.33/67.07 | 85.74/43.12 | 71.43/13.26 | 78.92/36.11 |
| Deep k-NN | 88.25/65.01 | 82.90/26.51 | 58.63/04.13 | 78.54/31.37 |
| ViM | 89.23/68.64 | 87.13/53.99 | 73.80/14.87 | 79.99/36.45 |
| IGEOOD | 90.11/66.43 | 86.81/44.56 | 71.41/04.17 | 77.45/35.34 |
| Standard FIM Trace (Ours) | 86.16/66.24 | 84.94/38.60 | 71.77/14.75 | 75.35/32.53 |
| Tensor FIM Trace (Ours) | 89.06/ 68.23 | 86.22/45.81 | 72.23/14.92 | 78.17/34.12 |
| Additive FIM Trace (Ours) | **91.28**/**68.88** | **87.23**/**54.47** | 74.38/**15.02** | 80.91/35.35 |

## 7 CONCLUSION

We introduced a novel perspective on OoD detection through the lens of Riemannian information geometry. Our work provides a unifying framework that first grounds common OoD signals in the framework of local Fisher-Rao sensitivity. More significantly, we extend this framework to provide a rigorous derivation for the additive structure. By formalizing the disentanglement of geometric components on a product manifold, we obtain a practical score that guarantees geometric invariance. Its competitive performance serves as strong validation of our orthogonality hypothesis. Beyond this specific score, our work highlights a larger paradigm shift in uncertainty quantification: moving from **global** approaches (e.g., Bayesian estimating posterior diversity (Fort et al., 2019; Lakshminarayanan et al., 2017)) to **local** geometric analysis. We demonstrate that uncertainty information is not only captured by the global exploration of the statistical manifold but is also encoded *locally*. Information geometry provides the language to leverage this local information efficiently. This work suggests that a deeper understanding of the geometric structures of our models is a critical step towards building more robust, and safe AI systems.

**Limitations and perspectives**. Our current work focuses on the geometric sensitivity within the parameter space of the final layer. While this provides a direct and analytic view of how features are mapped to predictions, OoD phenomena likely manifest in the geometry of deeper layers as well. Extending this sensitivity analysis to intermediate feature representations is a natural next step. Another opportunity for future work lies in the nature of the design choice for our additive score. While it is natural, motivated and justified by the geometric framework of product manifolds, its specific formulation is not unique. A direction for future work, would be to find a single, unified metric that intrinsically improves these geometric aspects.

## 8 REPRODUCIBILITY STATEMENT

All our experiments are conducted exclusively on publicly available datasets, such as CIFAR-10, CIFAR-100, SVHN, and ImageNet-1K. A detailed description of each dataset and its specific use in our evaluation protocol is provided in Appendix H. Our proposed algorithm is explicitly detailed throughout the paper.

For theoretical transparency, complete proofs for all our results are provided in Appendix E. We also include supplementary appendices that detail the underlying mathematical concepts in Appendix G and F, and provide comparisons with related work in Appendix C. Replicating our results is straightforward, as our methods are post-hoc and require no model retraining. To facilitate this process, we will release the complete source code for our experiments on GitHub upon publication shortly.

## 9 ETHICS

Our work is motivated by the need to create safer, frugal and more trustworthy AI systems. Our primary contribution is to improve the fundamental understanding of OoD behavior by using the theoretical lens of Information Geometry. We believe that a deeper understanding is essential for developing truly reliable machine learning methods. Note that our methods work without any exposure to OoD examples or modification of the forward process, either during training or at inference time. Our methods are designed to analyze existing pre-trained networks without requiring costly and energy-intensive retraining.

## ACKNOWLEDGMENTS

The authors would like to thank the anonymous reviewers for their constructive feedback and valuable comments that helped improve this manuscript. We acknowledge the support of the Thales company during this study for granting access to their facilities, materials and resources. Finally, A.N. wishes to thank Amandine for reviewing and proofreading the manuscript, and for her support.

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

# A    ABLATION STUDY

To validate our design choices and assess the robustness of our proposed method, we conduct a series of ablation studies using a ResNet-50 model on the ImageNet OoD benchmark. We investigate several aspects: the impact of the subspace construction methodology, the sensitivity of our method to the size of the calibration set and the hyperparameters for each component's contribution. In each study, we also report the performance of the individual geometric components of our score to understand their respective contributions.

## A.1    IMPACT OF SUBSPACE CONSTRUCTION STRATEGY

Our method's effectiveness rests on creating a discriminative subspace for the features. We compare our proposed supervised strategy, Linear Discriminant Analysis (LDA), against an unsupervised Principal Component Analysis (PCA) and a baseline using the identity matrix (i.e., no projection). The performance is evaluated using a fixed calibration set of 50000 samples.

Table 3 shows that the superiority of the LDA strategy is not just in the final combined score, but in the effectiveness of its geometric components.

With LDA, both the **Residual** and **Projected Norm** are strong OoD detectors on their own. In contrast, PCA fails to create a useful orthogonal complement; its Residual score is very weak (0.5892), indicating that the directions of maximal variance are not necessarily informative for OoD separation. Identity baseline: with no projection, the Residual signal is weak, confirming that our geometric decomposition is meaningful when applied to a well-chosen subdimensional subspace. This demonstrates that the supervised nature of LDA is important for creating a meaningful feature manifold where the orthogonal distance is a reliable OoD signal.

Table 3: **Ablation on Subspace Construction Strategy.** Performance breakdown on a ResNet-50. The supervised LDA strategy is unique in its ability to produce a highly discriminative Residual signal.

|  | **AUROC Contribution** | | |
| --- | --- | --- | --- |
| **Strategy** | **Residual** | **Proj. Norm** | **Combined** |
| Identity (No Proj.) | 50.00 | 61.07 | 78.16 |
| PCA | 58.92 | 60.35 | 76.73 |
| **LDA (Ours)** | **80.23** | **72.47** | **82.93** |

## A.2    SENSITIVITY TO CALIBRATION SET SIZE

Our method requires a set of ID samples to calibrate the subspaces. We evaluate the performance of our superior LDA-based approach while varying the size of the calibration set from 1,500 to 50,000 samples.

The results in Table 4 show that performance saturates quickly, confirming the method's practicality. The component-level breakdown offers a deeper insight: the **Residual** score is highly sensitive to the initial number of samples. With only 1,500 samples, its performance is poor (AUROC 38.62), suggesting that LDA cannot learn a generalizable ID manifold from such limited data. However, its effectiveness grows rapidly, becoming a strong signal by 2,500 samples and stabilizing after 10,000. The **Projected Norm**, while also improving, is less volatile.

Table 4: Sensitivity to calibration set size. AUROC performance using ResNet-50, reported as mean $\pm$ standard deviation over 3 random seeds. Performance improves with larger calibration sets, indicating robustness to the random selection

| Calib. Size | AUROC Contribution | | | |
| --- | --- | --- | --- | --- |
| | Uncertainty | Residual | Proj. Norm | Combined |
| 1,500 | $73.32 \pm 0.65$ | $37.78 \pm 0.88$ | $61.31 \pm 0.91$ | $76.04 \pm 0.76$ |
| 2,500 | $76.38 \pm 0.48$ | $77.73 \pm 0.76$ | $62.43 \pm 0.73$ | $81.12 \pm 0.65$ |
| 5,000 | $77.42 \pm 0.34$ | $79.06 \pm 0.43$ | $67.81 \pm 0.56$ | $82.00 \pm 0.32$ |
| 10,000 | $78.45 \pm 0.23$ | $79.96 \pm 0.25$ | $71.74 \pm 0.24$ | $82.21 \pm 0.26$ |
| 20,000 | $79.25 \pm 0.10$ | $80.19 \pm 0.14$ | $72.09 \pm 0.13$ | $82.99 \pm 0.11$ |
| 40,000 | $79.29 \pm 0.07$ | $81.25 \pm 0.09$ | $72.14 \pm 0.11$ | $82.91 \pm 0.08$ |
| 50,000 | $79.50 \pm 0.05$ | $81.46 \pm 0.06$ | $72.19 \pm 0.08$ | $82.97 \pm 0.04$ |

### A.3 SENSITIVITY TO $\lambda$ PARAMETERS

This figure shows the AUROC performance landscape for OoD detection. Performance is plotted against the magnitudes of two orthogonal feature components: the **Projected Norm** (how much a sample aligns with the learned ID patterns) and the **Residual** (the part of the sample that is orthogonal to those patterns).

The highest performance (the bright yellow diagonal band) is achieved when the projected norm is high and the Residual is high. This provides empirical validation for our decomposition in the feature space: that separating features into these orthogonal components provides a robust signal for OoD detection.

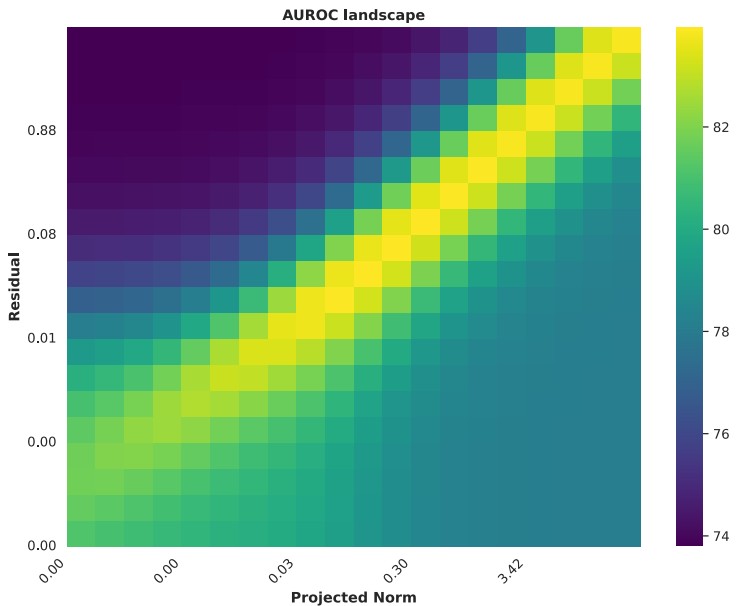

Figure 3: **AUROC Landscape for** $(|\lambda_M|, |\lambda_y|)$**.**

## B GEOMETRIC ASSUMPTION FOR ADDITIVITY VIA AN ORTHOGONALITY HYPOTHESIS ON THE PARAMETER MANIFOLD

This appendix provides a more intuitive geometric justification for the additive score in equation 13.

**The Orthogonality Hypothesis.** Our central hypothesis is that the parameter space $\mathcal{U}$ can be decomposed into two functionally orthogonal subspaces: one that primarily governs the geometric properties of the feature embeddings $f_{\hat{\theta}}(x)$, and another that primarily governs the mapping from features to the statistical manifold of predictions $p_{\hat{\theta}}(x)$. Intuitively, this orthogonal hypothesis implies the existence of distinct directions in the parameter space: those effectively perturbing the feature signal without altering the decision boundary, and vice-versa. Formally, we suppose that the tangent space at $\hat{\theta}$ decomposes into a direct sum of orthogonal subspaces $\mathcal{V}_f$ and $\mathcal{V}_p$:

$$T_{\hat{\theta}}\mathcal{U} \cong \mathcal{V}_f \oplus \mathcal{V}_p \tag{16}$$

such that the metric $g$ satisfies the orthogonality condition for any $u \in \mathcal{V}_f$ and $v \in \mathcal{V}_p$:

$$g_{\hat{\theta}}(u, v) = 0 \tag{17}$$

This orthogonality is motivated by empirical phenomena: for instance, a carefully crafted adversarial perturbation to an input $x$ can induce a large change in the output distribution $p_{\hat{\theta}}(x)$ while leaving the geometric properties of the feature vector $f_{\hat{\theta}}(x)$ nearly invariant.

**Formalizing the Hypothesis with a Composite Metric.** We can make this more precise by constructing a composite Riemannian metric tensor, $G_{total}$. Instead of using a single, monolithic Fisher metric, we suppose that $G_{total}$ as the concatenation of two distinct metric tensor in a block matrix, each designed to capture sensitivity along one of the orthogonal axes:

1. $G_p$: This metric tensor is constructed to measure sensitivity related to the **prediction space**. Its design is directly inspired by the original FIM, as this is the natural metric for a statistical manifold. Its trace, $\text{Tr}(G_p)$, is defined to be our uncertainty signal, $U_{\hat{\theta}}(x)$. It quantifies how parameter perturbations affect the output distribution $p_{\hat{\theta}}(x)$.

2. $G_f$: This metric tensor is constructed to measure sensitivity related to the **feature space**. It is designed to capture how parameter perturbations affect the geometric properties of the feature vector $f_{\hat{\theta}}(x)$. Its trace, $\text{Tr}(G_f)$, is defined as the weighted sum of our signals: the projected magnitude and the residual energy.

$$\text{Tr}(G_f) := \lambda_M \cdot M_{\hat{\theta}}(x) + \lambda_y \cdot \left( \|f_{\hat{\theta}}(x) - RR^\top f_{\hat{\theta}}(x)\|_2^2 \right).$$

The necessary mathematical structure to encode this orthogonality is a product manifold where each member is endowed wih $G_f, G_p$ respectively. We then naturally obtain our final score as the sum of these two components.

**The Additive Score as Total Sensitivity.** The total sensitivity on the product manifold is the trace of the total metric tensor $G_{total}$. By the linearity of the trace operator:

$$\text{Tr}(G_{total}) = \text{Tr}(G_p) + \text{Tr}(G_f). \tag{18}$$

Substituting the definitions for the traces of our component metrics, we directly recover the final additive score:

$$\text{Tr}(G_{total}) = \underbrace{U_{\hat{\theta}}(x)}_{\text{Prediction Sensitivity}} + \underbrace{\lambda_M M_{\hat{\theta}}(x) + \lambda_y \left( \|f_{\hat{\theta}}(x) - RR^\top f_{\hat{\theta}}(x)\|_2^2 \right)}_{\text{Feature Sensitivity}}. \tag{19}$$

The additive score is interpreted as the total sensitivity of a parameter manifold equipped with a composite metric. Thus, this metric is explicitly constructed to reflect our plausible hypothesis that OoD phenomena manifest along two independent orthogonal geometric axes: one governing the feature space and one governing the prediction space.

# C  DETAILED COMPARISON WITH RELATED GRADIENT AND INFORMATION GEOMETRY-BASED METHODS

This appendix provides a detailed comparison between our Fisher-Rao sensitivity approach (specifically the Standard FIM Trace) and two closely related methods: GradNorm (Huang et al., 2021), which also utilizes gradient information, and IGEOOD (Gomes et al., 2022), which also leverages information geometry.

## C.1  COMPARISON WITH GRADNORM

GradNorm proposes an OoD detection score based on the vector norm of gradients backpropagated from a specific loss function. While both GradNorm and our Standard FIM Trace utilize gradient information and offer closed-form solutions for the last layer, their conceptual foundations and mathematical formulations are different.

The Standard FIM Trace measures the local sensitivity of the model $p(y|x, \theta)$ to parameter perturbations around the trained weights $\hat{\theta}$. It quantifies the magnitude of the Fisher-Rao metric tensor locally, acting as a proxy for the stability of the optimization landscape at that point for the input $x$.

GradNorm is motivated by optimization dynamics. It measures the magnitude of the gradient of the Kullback-Leibler divergence between the model's softmax output and a fixed uniform distribution $u$: $\mathcal{L}_{Uni}(x) = D_{KL}(u||p(\cdot|x, \hat{\theta}))$. The score is $S_{GradNorm}(x) = ||\nabla_\theta \mathcal{L}_{Uni}(x)||_p$. Huang et al. hypothesize that ID inputs, being less uniform, have a larger gradient magnitude towards uniformity.

### C.1.1  MATHEMATICAL SETTING

The fundamental difference lies in how the gradients are computed and aggregated. Let $W$ be the last-layer weights and $p(\cdot|x, \hat{\theta})$ the softmax output for input $x$.

The **Standard FIM Trace** (equation 7) is defined as:

$$\text{Tr}(F_{W,x}(\hat{\theta})) = \mathbb{E}_{y \sim p_{\hat{\theta}}(x)}[||\nabla_W \log p(y|x, \hat{\theta})||_2^2]. \tag{20}$$

This is the expectation of the squared L2 norm of the gradients, where the expectation is taken with respect to the model's own predictive distribution $p_{\hat{\theta}}(x)$.

The gradient used in GradNorm is equivalent to the average of the gradients of the cross-entropy loss for all labels:

$$\nabla_W D_{KL}(u||p_{\hat{\theta}}(x)) = \frac{1}{K} \sum_{k=1}^{K} \nabla_W \mathcal{L}_{CE}(p_{\hat{\theta}}(x), k) = \mathbb{E}_{y \sim u}[\nabla_W \log p(y|x, \hat{\theta})]. \tag{21}$$

If we consider the L2 norm for a comparison:

$$S_{GradNorm,L2}(x) = ||\mathbb{E}_{y \sim u}[\nabla_W \log p(y|x, \hat{\theta})]||_2^2. \tag{22}$$

This is the squared L2 norm of the expectation of the gradients, where the expectation is taken with respect to the uniform distribution $u$.

Hence, we have the following key distinctions:

1. **Jensen Inequality:** The FIM Trace calculates $E[||g||^2]$ (the second moment), while GradNorm calculates $||E[g]||^2$. By Jensen's inequality, $E[||g||^2] \geq ||E[g]||^2$. The FIM Trace captures the variance of the gradients, whereas GradNorm captures the magnitude of the mean gradient direction.

2. **Base Distribution:** The FIM Trace is intrinsic to the model's geometry defined by $p(\cdot|x, \theta)$, while GradNorm depends on an external reference distribution $u$.

### C.1.2  CLOSED-FORM FACTORIZATION

Both methods yield a closed-form factorization when restricted to the last layer. Let $f_{\hat{\theta}}(x)$ be the feature extractor output.

- **Standard FIM Trace:** $||f_{\hat{\theta}}(x)||_2^2 \cdot (1 - \sum_k p_{k,\hat{\theta}}(x)^2)$.
- **GradNorm (L1, optimal variant):** $||f_{\hat{\theta}}(x)||_1 \cdot (\sum_k |1 - K \cdot p_{k,\hat{\theta}}(x)|)$.

While both capture similar intuitions (feature magnitude and output uncertainty), the specific metrics used ($L_2^2$ vs $L_1$ for features; Gini impurity vs. L1 deviation from uniform for outputs) lead to different behaviors.

## C.2 Comparison with IGEOOD

IGEOOD also utilizes information geometry tools, specifically the Fisher-Rao distance, for OOD detection.

### C.2.1 Local Sensitivity vs. Geodesic Distance

The fundamental difference is the distinction between measuring the magnitude of the metric tensor (local sensitivity) and measuring the distance along the manifold induced by that metric (geodesic distance).

**FIM Trace:** We analyze the geometry locally around the trained parameters $\hat{\theta}$. The FIM Trace, $\mathrm{Tr}(F_x(\hat{\theta}))$, is a scalar summary of the FIM $F_x(\hat{\theta})$, which acts as the Riemannian metric tensor at $\hat{\theta}$. It quantifies how rapidly the statistical distance $ds^2$ increases due to infinitesimal parameter changes $d\theta$.

**IGEOOD:** IGEOOD utilizes the Fisher-Rao distance, which is the geodesic distance on the statistical manifold. It measures the shortest path between two distinct distributions on the manifold. IGEOOD applies this by comparing the distribution associated with a test input $x$ to distributions derived from the training data.

### C.2.2 Methodology

**FIM Trace:** Our method is characterized by a focus on the trained model's stability at $\hat{\theta}$. It provides computationally efficient, forward-only scores that do not require comparison against stored training data statistics during inference.

**IGEOOD:** IGEOOD is essentially a distance-based detector (akin to Mahalanobis distance or KNN methods) but utilizes the intrinsic Riemannian distance. This involves:

1. Computing class prototypes from the training data. At the logits, this involves optimizing centroids in the Fisher-Rao geometry. At the feature space $f_{\hat{\theta}}(x)$, it involves estimating parameters for Gaussian models.

2. During inference, computing the geodesic distance between the test sample's representation and these stored centroids.

3. The final score is the Rao distance to the nearest centroid or a sum of distances.

While both methods are grounded in information geometry, our approach focuses on the intrinsic local stability of the trained model for a given input, whereas IGEOOD focuses on measuring the geodesic distance between the input and the learned distribution of the training data.

## C.3 Comparison with ViM

ViM (Wang et al., 2022) is a highly relevant baseline, sharing a fundamental insight with our work: the necessity of combining class-dependent information with information from the feature space. Both methods identify that only relying on one source is potentially not enough for strong OoD detection. ViM addresses this by constructing a virtual logit from the feature residual, expanding the class set to $K + 1$. While the resulting signals are similar, the motivations and derivations mechanisms differ:

1. ViM depends on a construction where the norm of the residual $||x^{P^\perp}||$ is treated as a new logit. The final score is the softmax probability of this class. Our Additive FIM Trace

is derived from the local sensitivity of the statistical manifold. The additive structure is a theoretical consequence of modeling the total sensitivity on a product manifold, where the prediction space and the orthogonal feature space are treated as independent geometric components.

2. To combine the residual with the original logits, ViM introduces a scaling factor $\alpha$, determined to match the average magnitude of the maximum logits on the training set. This acts as a normalization constant to make the virtual logit comparable. Conversely, our hyperparameters $\lambda_M$ and $\lambda_y$ comes from the principle of indifference, applied to the variances of the sensitivity components. Our aim is to equilibrate the contribution of each geometric axis to the total variance of the score, under a statistical independence assumption.

3. ViM use PCA on the feature space to define the residual. Our framework employs a Tensor Factorization of the FIM, which motivates the use of LDA for the feature subspace $R$ and PCA for the probability subspace $Q$.

Hence, ViM can be viewed as an effective application of the principle of combining feature residuals and logits. Our work provides a formal geometric framework for this high-level idea.

## D  ANALYSIS OF THE STANDARD AND TENSOR FIM TRACE

In this section, we analyze the limitations of the FIM Trace scores derived from the product formulation. Although we focus on the Standard FIM Trace for simplicity, the reasoning applies to the Tensor FIM Trace, as both share the same multiplicative structure.

The Standard FIM Trace is defined as $\mathrm{Tr}(F_{W,x}(\hat{\theta})) = \|f_{\hat{\theta}}(x)\|_2^2(1 - \|p_{\hat{\theta}}(x)\|_2^2)$. This equation shows that sensitivity is the product of two factors:

1. **Magnitude:** The magnitude of the feature representation ($\|f_{\hat{\theta}}(x)\|_2^2$).

2. **Uncertainty:** The flatness of the softmax prediction ($1 - \|p_{\hat{\theta}}(x)\|_2^2$).

For this score to detect OoD data, we want the product to be higher for OoD than for ID. However, research suggests these two factors often conflict: Haas et al. (2023) noted that modern architecture learns features of similar magnitude for both ID and OoD inputs (a phenomenon related to Neural Collapse). If the magnitudes are similar, the score relies only on uncertainty, which might not be enough for good separation. More importantly, alternative studies (Sun et al., 2021; Ammar et al., 2023; Nguyen et al., 2025) found that OoD inputs generally have lower feature magnitudes than ID inputs. This creates a problem for the product formulation:

- ID data tends to have High Magnitude and Low Uncertainty.
- OoD data tends to have Low Magnitude and High Uncertainty.

Because the Standard FIM Trace and Tensor FIM Trace multiply these factors, the lower magnitude of OoD data counteracts the effect of its higher uncertainty. This makes the product score suboptimal for OoD detection.

The following result formalizes this tension between magnitude and uncertainty in the product formulation:

**Corollary D.1** (Separation conditions for the FIM Trace product). *Let the Standard FIM Trace be* $S(x) = \|f_{\hat{\theta}}(x)\|_2^2(1 - \|p_{\hat{\theta}}(x)\|_2^2)$. *Assume bounds on the feature magnitude:*

$$\|f_{\hat{\theta}}(x)\|_2^2 \in [m_{\mathrm{ID}}, M_{\mathrm{ID}}] \quad for\ x \in \mathcal{X}_{\mathrm{ID}}, \qquad \|f_{\hat{\theta}}(x)\|_2^2 \in [m_{\mathrm{OoD}}, M_{\mathrm{OoD}}] \quad for\ x \in \mathcal{X}_{\mathrm{OoD}}. \quad (23)$$

*We assume OoD has smaller norms, so* $M_{\mathrm{OoD}} \le M_{\mathrm{ID}}$ *(and typically* $m_{\mathrm{OoD}} \le m_{\mathrm{ID}}$*).*

*Assume bounds on the uncertainty (flatness):*

$$1 - \|p_{\hat{\theta}}(x)\|_2^2 \le \varepsilon_{\mathrm{ID}} \quad \forall x \in \mathcal{X}_{\mathrm{ID}}, \qquad 1 - \|p_{\hat{\theta}}(x)\|_2^2 \ge \varepsilon_{\mathrm{OoD}} \quad \forall x \in \mathcal{X}_{\mathrm{OoD}}. \quad (24)$$

*We assume OoD is flatter, so* $\varepsilon_{\mathrm{OoD}} > \varepsilon_{\mathrm{ID}}$ *(and* $m_{\mathrm{OoD}} > 0$*).*

*Then the bounds on the score* $S(x)$ *are:*

$$\max_{x \in \mathcal{X}_{\mathrm{ID}}} S(x) \le M_{\mathrm{ID}}\,\varepsilon_{\mathrm{ID}}, \qquad \min_{x \in \mathcal{X}_{\mathrm{OoD}}} S(x) \ge m_{\mathrm{OoD}}\,\varepsilon_{\mathrm{OoD}}. \quad (25)$$

*For **strict separation**, the uncertainty gain must outweigh the magnitude loss. Specifically:*

$$\frac{\varepsilon_{\text{OoD}}}{\varepsilon_{\text{ID}}} \;>\; \frac{M_{\text{ID}}}{m_{\text{OoD}}}. \tag{26}$$

*If this condition holds, then $\min_{x \in \mathcal{X}_{\text{OoD}}} S(x) > \max_{x \in \mathcal{X}_{\text{ID}}} S(x)$.*

### D.1 Visualization of the Magnitude-Uncertainty Trade-off

To validate the theoretical separation conditions discussed in Corollary D.1 and motivate the additive decomposition, we analyze the geometric signal dynamics in a controlled environment.

#### D.1.1 Experimental Setup for Geometric Visualization

To isolate the intrinsic behavior of Fisher-Rao sensitivity without the under-performance of the norm signal induced by complex architectures in large-scale benchmarks, we adopt a toy experimental setting. **Datasets.** We use a standard digit recognition setup for OoD analysis:

- **In-Distribution (ID):** The **MNIST** dataset.
- **Out-of-Distribution (OoD):** The **SVHN** dataset.

**Model Configuration.** The architecture consists of a simple MLP with two hidden layers: 512 and 128 units, ReLU activations. The penultimate layer produces a 128-dimensional feature representation $f_{\hat{\theta}}(x)$, which is then fully connected to 10 class logits. Optimization is performed using SGD to minimize the standard CE Loss.

#### D.1.2 Analysis of Signal Dynamics

Figure 4 illustrates the joint distribution of feature magnitude ($M_{\hat{\theta}}(x) = \|f_{\hat{\theta}}(x)\|_2^2$) and predictive uncertainty ($U_{\hat{\theta}}(x) = 1 - \|p_{\hat{\theta}}(x)\|_2^2$).

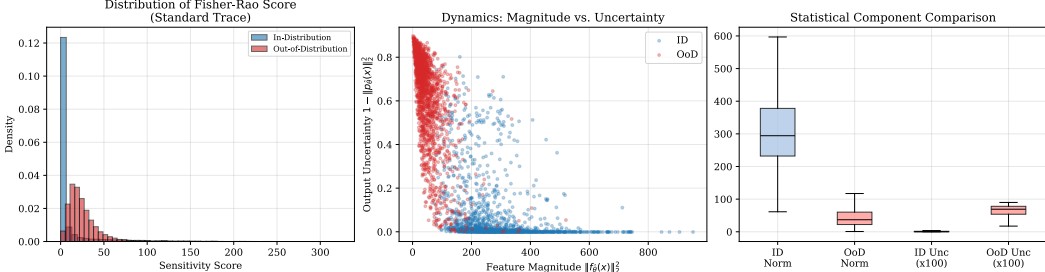

Figure 4: Visualization of the Geometric Decomposition. **Left:** Histogram of the Standard FIM Trace scores showing the imperfect separation between ID (blue) and OoD (red). **Middle:** Scatter plot of Feature Magnitude vs. Output Uncertainty. We observe distinct geometric regimes: ID data is clustered in the high-magnitude/low-uncertainty region, while OoD data shifts to the high-uncertainty/low-magnitude region. **Right:** Box plots showing the collapse of feature magnitude for OoD data, which dominates the multiplicative score.

**Why is ID Sensitivity Lower?** The scatter plot (middle panel) provides a clear geometric explanation for why the local Fisher-Rao sensitivity is consistently lower for ID data: ID samples (blue points) are concentrated at the very bottom of the uncertainty axis. Since the Standard FIM Trace is a multiplicative score, this quasi-zero uncertainty acts as an anchor, forcing the total sensitivity to remain low regardless of the high feature magnitude. For OoD samples (red points), the uncertainty increases, driving the sensitivity up. However, this effect is counteracted by the simultaneous collapse of feature magnitude visible in the box plots on the right. This empirical evidence justifies the decoupling by moving to an additive product manifold, preventing the magnitude signal from suppressing the strong uncertainty signal.

# E    PROOFS OF THEORETICAL RESULTS

*Theorem 5.1.* For the softmax last layer, the gradient of the log-probability w.r.t. a weight $W_{k,i}$ is

$$\frac{\partial \log p(y|x,\theta)}{\partial W_{k,i}} = \left(\mathbf{1}\{y{=}k\} - p_{k,\theta}(x)\right) f_i(x). \tag{27}$$

Taking the expectation over $y \sim p(\cdot \mid x, \theta)$,

$$\left(\operatorname{diag} F_{W,x}\right)_{k,i} = \mathbb{E}_y\left[\left(\frac{\partial \log p(y|x,\theta)}{\partial W_{k,i}}\right)^2\right] = f_{i,\theta}(x)^2 \, p_{k,\theta}(x)\left(1 - p_{k,\theta}(x)\right). \tag{28}$$

Summing over all $(k,i)$ gives the closed-form trace:

$$\operatorname{Tr}\!\left(F_{W,x}(\theta)\right) = \|f_\theta(x)\|_2^2 \left(1 - \|p_\theta(x)\|_2^2\right). \tag{29}$$

$\square$

*Proposition 5.2.* Fix an input $x$ and $\theta \in \mathbb{R}^m$. Let $f_\theta(x) \in \mathbb{R}^D$ be the penultimate representation and $p_{\hat{\theta}}(x) \in \mathbb{R}^K$ the softmax probabilities with entries $p_{k,\hat{\theta}}(x) = p(y = k|x, \theta)$. For the last linear layer $z = W^\top f_{\hat{\theta}}(x) + b$ and $p = \operatorname{softmax}(z)$, the gradient of the log-probability with respect to a weight $W_{k,i}$ is

$$\frac{\partial \log p(y|x,\theta)}{\partial W_{k,i}} = \left(\mathbf{1}\{y{=}k\} - p_{k,\hat{\theta}}(x)\right) f_i(x). \tag{30}$$

Vectorizing the last-layer weights $W \in \mathbb{R}^{D \times K}$, the per-class gradient can be written as

$$g_y(x) := \operatorname{vec}\!\left(\nabla_W \log p(y|x,\theta)\right) = \left(e_y - p_{\hat{\theta}}(x)\right) \otimes f_{\hat{\theta}}(x), \tag{31}$$

where $e_y$ is the $y$-th canonical basis vector in $\mathbb{R}^K$ and $\otimes$ denotes the tensor product.

The input-wise Fisher block for $W$ is the second moment of this score under $y \sim p(\cdot \mid x; \theta)$:

$$F_{W,x} = \mathbb{E}_y\!\left[g_y(x)\,g_y(x)^\top\right] = \left(\mathbb{E}_y\!\left[(e_y - p)(e_y - p)^\top\right]\right) \otimes \left(f_\theta(x) f_\theta(x)^\top\right). \tag{32}$$

Since $\mathbb{E}_y[e_y e_y^\top] = \operatorname{diag}(p_{\hat{\theta}}(x))$ and $\mathbb{E}_y[e_y] = p_{\hat{\theta}}(x)$, we have

$$\mathbb{E}_y\!\left[(e_y - p)(e_y - p)^\top\right] = \operatorname{diag}(p_{\hat{\theta}}(x)) - p_{\hat{\theta}}(x) p_{\hat{\theta}}(x)^\top =: S_{p_{\hat{\theta}}(x)}. \tag{33}$$

Let $Q \in \mathbb{R}^{K \times r_c}$ and $R \in \mathbb{R}^{D \times r_f}$, and set $P := Q \otimes R \in \mathbb{R}^{(DK) \times (r_c r_f)}$. Using the mixed-product property of the tensor product,

$$(Q \otimes R)^\top (A \otimes B)(Q \otimes R) = (Q^\top A Q) \otimes (R^\top B R), \tag{34}$$

we obtain

$$P^\top F_{W,x} P = (Q^\top S_{p_{\hat{\theta}}(x)} Q) \otimes (R^\top f_{\hat{\theta}}(x) f_{\hat{\theta}}(x)^\top R). \tag{35}$$

Taking traces and using $\operatorname{Tr}(A \otimes B) = \operatorname{Tr}(A)\operatorname{Tr}(B)$ yields

$$\operatorname{Tr}\!\left(P^\top F_{W,x} P\right) = \operatorname{Tr}\!\left(Q^\top S_{p_{\hat{\theta}}(x)} Q\right) \cdot \operatorname{Tr}\!\left(R^\top f_{\hat{\theta}}(x) f_{\hat{\theta}}(x)^\top R\right). \tag{36}$$

Finally, since $R^\top f_{\hat{\theta}}(x) f_{\hat{\theta}}(x)^\top R = (R^\top f_{\hat{\theta}}(x))(R^\top f_{\hat{\theta}}(x))^\top$ is rank-1,

$$\operatorname{Tr}\!\left(R^\top f_\theta(x) f_\theta(x)^\top R\right) = \|R^\top f_\theta(x)\|_2^2, \tag{37}$$

which proves the proposition. $\square$

*Proposition 5.3.* Let be $S$ the additive score. Its developed expression for the variance is:

$$\begin{aligned}
\operatorname{Var}(S) = \ &\operatorname{Var}(U) + \lambda_y^2 \operatorname{Var}(y) + \lambda_M^2 \operatorname{Var}(M) \\
&+ 2\lambda_y \operatorname{Cov}(U, y) + 2\lambda_M \operatorname{Cov}(U, M) + 2\lambda_y \lambda_M \operatorname{Cov}(y, M)
\end{aligned}$$

Under the independence assumption, the covariance terms vanish. Then, the global variance boils down to:

$$\operatorname{Var}(S) = \operatorname{Var}(U) + \lambda_y^2 \operatorname{Var}(y) + \lambda_M^2 \operatorname{Var}(M) \tag{38}$$

To ensure that each component contributes equally to this global variance, one must satisfy:

$$\lambda_y^2 \operatorname{Var}(y) = \operatorname{Var}(U) \quad \text{and} \quad \lambda_M^2 \operatorname{Var}(M) = \operatorname{Var}(U). \tag{39}$$

This leads to the exact same analytical solution as minimizing $\mathcal{L}$. Thus, the statistical independence assumption guarantees that minimizing the simple marginal error $\mathcal{L}$ effectively balances the total variance of the score. $\square$

*Proposition 5.4.* Point 1 and 2 are trivial. We details here the proof for the uniform scaling. Recall that:

$$\lambda_y = \sqrt{\frac{\text{Var}(U_{\hat\theta})}{\text{Var}(y_{\hat\theta})}}, \quad \lambda_M = \sqrt{\frac{\text{Var}(U_{\hat\theta})}{\text{Var}(M_{\hat\theta})}} \tag{40}$$

up to a sign. Consider the rescaling reparameterization: $f'_{\hat\theta} = cf_{\hat\theta}$, where $c \in \mathbb{R}^*$, assuming the classifier weights adapt ($W' = W/c$) to maintain invariant predictions.

1. The uncertainty $U_{\hat\theta}$ (based on probabilities) remains invariant: $U' = U$.

2. The embedding-based terms (squared norms) scale quadratically: $M'_{\hat\theta} = \|cf_{\hat\theta}\|^2 = c^2 M_{\hat\theta}$ and $y'_{\hat\theta} = c^2 y_{\hat\theta}$.

3. Consequently, their variances scale by $c^4$: $\text{Var}(M'_{\hat\theta}) = c^4 \text{Var}(M_{\hat\theta})$.

If the $\lambda_M$ hyperparameter value were computed on the rescaled data, it would adapt as follows:

$$\lambda'_M = \sqrt{\frac{\text{Var}(U'_{\hat\theta})}{\text{Var}(M'_{\hat\theta})}} = \sqrt{\frac{\text{Var}(U_{\hat\theta})}{c^4 \text{Var}(M_{\hat\theta})}} = \frac{1}{c^2}\lambda_M \tag{41}$$

The scaling factor $c^2$ from the feature magnitude is canceled by the $1/c^2$ scaling of the hyperparameter:

$$\lambda'_M M'_{\hat\theta} = \left(\frac{1}{c^2}\lambda_M\right)(c^2 M_{\hat\theta}) = \lambda_M M_{\hat\theta} \tag{42}$$

The same applies to the residual term. $\qquad\square$

## F  TANGENT SPACE, RIEMANNIAN CURVATURE AND CONFORMAL TRANSFORMATION

### F.1  THE FISHER-RAO METRIC AND THE TANGENT SPACE

Consider a statistical manifold $S = \{p(\cdot, \theta) \mid \theta \in \Theta\}$. Information geometry endows $S$ with the FIM $g = F(\theta)$ as its natural (pseudo)-Riemannian metric tensor. The components are given by $g_{ij}(\theta) = F_{ij}(\theta)$.

We start by fixing a local coordinate basis $\{d\theta_1, \ldots, d\theta_p\}$ of the tangent space $T_\theta(S)$. The Fisher Information Matrix acts as a (pseudo)-Riemannian metric on this space, defined by

$$\langle d\theta_i, d\theta_j \rangle = g_{ij}(\theta). \tag{43}$$

where $\langle , \rangle$ is strictly defined positive or negative. In particular, the squared length of each basis vector is $\|d\theta_i\|^2 = g_{ii}(\theta)$. Summing these contributions gives

$$\text{Tr}(F(\theta)) = \sum_{i=1}^{p} \|d\theta_i\|^2.$$

Thus, the trace aggregates the metric lengths of an orthonormal coordinate frame, and in some sense, measures the "size" of the tangent space. This motivate us to use it as a geometric invariant, quantifying the uncertainty across all directions of the parameter space.

### F.2  RIEMANNIAN CURVATURE

Riemannian curvature measures the extent to which the manifold deviates from being locally Euclidean.

Formally, curvature involves the derivatives of the metric components, $\partial g_{ij}/\partial \theta_k$. These derivatives define the connection on the manifold, often represented by the Christoffel symbols (Gallot et al., 1990). The geometric structure is described by the Riemann curvature tensor, which is a rank-4 tensor derived from the Christoffel symbols and their derivatives. Contractions of this tensor yield the Ricci tensor and the Scalar curvature. The distinction is:

1. **The Metric** $F(\theta)$**:** Measures the magnitude of the sensitivity at a specific point $\theta$. It is a first-order geometric property.

2. **The Curvature** $R(\theta)$**:** Measures the rate of change of the metric in the neighborhood of $\theta$. It is a higher-order geometric property.

## F.3 CONFORMAL TRANSFORMATIONS AND PRODUCT MANIFOLDS

**Definition F.1** (Conformal and Pseudo-Conformal Transformation). *Let $(M, g)$ be a Riemannian manifold with metric tensor $g$. A transformation of $g$ is the construction of a new metric*

$$g' = \varphi \cdot g,$$

*where $\varphi : M \to I$ is a smooth function.*
*If $I = ]0, +\infty[$ we say that the transformation is conformal. If $I = ] - \infty, 0[$, we say that the transformation is pseudo-conformal. In the case where $I = ] - \infty, 0[$, $g'$ is a peudo-Riemannian metric where the notion of curvature and local sensitivity is well defined (see Appendix F).*

**Product Manifold.** Given $n$ Riemannian manifolds $(M_1, g_1), (M_2, g_2),...,(M_n, g_n)$, their product manifold is the Cartesian product $M = M_1 \times M_2, ... \times M_n$ equipped with the natural product Riemannian metric

$$g\big((v_1, v_2, ..., v_n), (w_1, w_2, ..., w_n)\big) = g_1(v_1, w_1) + g_2(v_2, w_2) + ... + g_n(v_n, w_n)$$

for tangent vectors $(v_1, v_2, ..., v_n), (w_1, w_2, ...w_n) \in T_{(x_1, x_2, ..., x_n)}M$.

Now for each $i$, fix a basis $\{v_1^{(i)}, \ldots, v_{n_i}^{(i)}\}$ of the tangent space $T_{x_i}M_i$. The metric tensor of $M$ in the basis $\{(v_1^{(1)}, 0, \ldots, 0), ...(v_{k_1}^{(1)}, 0, \ldots, 0), \ldots (0, ...v^{(n)_{k_n}})\}$ of $T_{(x_1, ..., x_k)}M$ takes the block-diagonal form

$$G_M = \begin{pmatrix} \big(g_1(v_p^{(1)}, v_q^{(1)})\big)_{p,q=1,...,k_1} & 0 & \cdots & 0 \\ 0 & \big(g_2(v_p^{(2)}, v_q^{(2)})\big)_{p,q=1,...,k_2} & \cdots & 0 \\ \vdots & \vdots & \ddots & \vdots \\ 0 & 0 & \cdots & \big(g_n(e_p^{(n)}, e_q^{(n)})\big)_{p,q=1,...,k_n} \end{pmatrix}.$$

Each block represent a single metric $g_i$ on a manifold $M_i$

# G SOME PROPERTIES OF THE TENSOR PRODUCT

Section 5.2.1 utilizes the tensor product to factorize the FIM of the last layer and to define restricted subspaces. This appendix reviews the definition and the key properties used in the derivations for completeness.

## G.1 DEFINITION AND VECTORIZATION

**Definition (Tensor Product (Bourbaki, 2006)).** For matrices $A \in \mathbb{R}^{a \times b}$ and $B \in \mathbb{R}^{c \times d}$, the Tensor product $A \otimes B \in \mathbb{R}^{ac \times bd}$ is defined as the block matrix:

$$A \otimes B = \begin{bmatrix} a_{11}B & \cdots & a_{1b}B \\ \vdots & \ddots & \vdots \\ a_{a1}B & \cdots & a_{ab}B \end{bmatrix}. \tag{44}$$

The Tensor product is closely related to the vectorization operator, $vec(\cdot)$, which stacks the columns of a matrix. For vectors $u$ and $v$:

$$vec(vu^\top) = u \otimes v. \tag{45}$$

This property is used to express the vectorized gradient of the log-probability with respect to the last-layer weights $W$:

$$vec(\nabla_W \log p(y|x; \theta)) = (e_y - p_{\hat{\theta}}(x)) \otimes f_{\hat{\theta}}(x). \tag{46}$$

Indeed, recall that the last layer produces logits $z = W^\top f \in \mathbb{R}^K$, with softmax probabilities $p = \text{softmax}(z)$ and one-hot vector $e_y \in \mathbb{R}^K$ for the label $y$. We use the chain rule on the logits to get:

$$\frac{\partial \log p_y}{\partial z_k} = \mathbf{1}\{k = y\} - p_k = (e_y - p)_k. \tag{47}$$

And by simple derivation we have:

$$\frac{\partial z_k}{\partial W} = f\, e_k^\top, \tag{48}$$

which is a matrix of size $D \times K$ with the $k$-th column equal to $f$. So the gradient is simply:

$$\nabla_W \log p_y = \sum_{k=1}^K (e_y - p)_k\, f\, e_k^\top = f\,(e_y - p)^\top. \tag{49}$$

Finally, by the vectorization identity $\text{vec}(ab^\top) = b \otimes a$, we obtain

$$\text{vec}(\nabla_W \log p_y) = (e_y - p) \otimes f. \tag{50}$$

### G.2 Key Algebraic Properties

The derivations rely on the following algebraic properties:

**Proposition G.1.** *1. Mixed-Product Property: If the matrix dimensions are compatible for the products $AC$ and $BD$ to exist, then:*

$$(A \otimes B)(C \otimes D) = (AC) \otimes (BD). \tag{51}$$

***2. Transpose Property:***

$$(A \otimes B)^\top = A^\top \otimes B^\top. \tag{52}$$

## H Details on training and dataset

### H.1 Experimental Details

For experiments on CIFAR-10 and CIFAR-100, we trained a ResNet-18 architecture from scratch for 200 epochs. The optimization was performed using SGD with Nesterov momentum (0.9), a weight decay of $5 \times 10^{-4}$, and a batch size of 256. We employed a cosine annealing learning rate schedule defined as:

$$\eta_t = \eta_{\min} + \tfrac{1}{2}(\eta_0 - \eta_{\min})(1 + \cos(\pi t / T_{\max})),$$

with $\eta_0 = 0.1$, $\eta_{\min} = 0$, and $T_{\max} = 200$. The training objective was the standard cross-entropy loss. Input data was normalized using the following channel-wise statistics:

- **CIFAR-10:** $\mu = (0.4914, 0.4822, 0.4465)$, $\sigma = (0.2470, 0.2435, 0.2616)$.
- **CIFAR-100:** $\mu = (0.5070, 0.4865, 0.4409)$, $\sigma = (0.2673, 0.2564, 0.2761)$.

### H.2 Dataset Details

**CIFAR-10** CIFAR-10 (Krizhevsky et al., 2009) consists of 10 object classes and contains a training set with 50,000 images and a testing set including 10,000 images. Each of the images is RGB and of size 32×32 pixels

**CIFAR-100** CIFAR-100 (Krizhevsky et al., 2009) consists of 100 object classes and contains a training set with 50,000 images and a testing set including 10,000 images. Each of the images is RGB and of size 32×32 pixels

**ImageNet-1K**   Large Scale Visual Recognition Challenge (ILSVRC) 2012 dataset (Deng et al., 2009), commonly referred to as ImageNet-1K, is a large-scale benchmark for object recognition. It contains 1,000 diverse object classes, with approximately 1.28 million images in its training set and 50,000 images in the validation set. The images are high-resolution color photographs, which are typically resized to 224×224 pixels before being fed into a model.

**SVHN**   The Street View House Numbers (SVHN) dataset (Netzer et al., 2011) is a large-scale

dataset of 600,000 images of digits obtained from house numbers in Google Street View. In our work, we kept a fixed set of 10,000 images that we resized to 32×32.

**ImageNet-O**   The ImageNet-Outlier dataset (Hendrycks et al., 2021) is a test set specifically designed for OoD detection for models trained on ImageNet-1K. It consists of images that do not belong to any of the original 1,000 ImageNet-1K classes. It was created to be a challenging benchmark testing a model's understanding of its learned semantic concepts

**Places365**   The Places365 dataset (Zhou et al., 2017) is a large-scale scene recognition benchmark. Unlike object-centric datasets like ImageNet, its primary goal is to train models that can classify the environment or context of an image (e.g., "bakery," "coast," "forest path"). It contains over 1.8 million images categorized into 365 distinct scene classes.

**iNaturalist**   The iNaturalist (Van Horn et al., 2018) dataset is a large collection of photos of plants and animals from all over the world. It is created by a community of users on the iNaturalist platform. Because it includes many different species, it is often used to test a model's ability to perform visual classification on the difference between very similar categories, like different species of birds or insects.

**SUN**   The SUN (Scene UNderstanding) dataset (Xiao et al., 2010) is a collection of images designed for training models to recognize different environments or scenes. The SUN dataset's goal is to classify the entire context of an image, such as "kitchen," "forest," "beach," or "office." It contains a wide variety of indoor and outdoor scenes.

