# OpenReview forum: "Fisher-Rao Sensitivity for Out-of-Distribution Detection in Deep Neural Networks"
_ICLR.cc/2026/Conference — ICLR 2026 Poster_

### Official Review · Reviewer_LmQv · 2025-10-24

**Soundness:** 2
**Presentation:** 3
**Contribution:** 2
**Rating:** 4
**Confidence:** 4

**Summary:**

The paper approaches the OOD detection problem from the perspective of information geometry, using the Fisher–Rao metric. It posits that the *per-sample* trace of the Fisher Information Matrix (FIM) tends to be larger for OOD samples than for in-distribution (ID) ones, meaning that small parameter perturbations cause greater output changes on OOD points. The FIM trace is also interpreted as the product of a "softmax uncertainty" term and the magnitude of the penultimate-layer features. Building on this view, the authors propose three OOD detection methods: (1) Standard FIM Trace, using the full last-layer parameter space; (2) Tensor FIM Trace, which projects the two multiplicative components onto subspaces learned from ID data; and (3) Additive FIM Trace, which replaces the product with a weighted sum of these terms and adds a third component capturing the penumltimate features magnitude orthogonal to the learned ID subspace.

**Strengths:**

**(S1) Presentation and writing:** The paper is clearly written and well-structured. The Fisher Information framework for OOD detection is effectively introduced and its connection to the first two proposed metrics is well motivated (though the link to the third metric remains less clear; see W1).

**(S2) Analysis of FIM trace metric:** The work offers an interesting perspective on the FIM trace with respect to the penultimate layer parameters, interpreting it as the product of a softmax uncertainty term and the norm of the penultimate layer features. This provides a way to understand the factors influencing OOD detection performance in this and prior works.

**Weaknesses:**

**(W1) Motivation for the additive FIM trace score:** My main concern is that the motivation and "derivation" of the additive FIM trace score do not align with the Fisher–Rao framework developed earlier in the paper. The additive score is introduced by defining a metric on a product of three manifolds, but this construction is no longer Fisher–Rao, lacks a clear statistical interpretation, and is not derived from information-geometric principles. As a result, the proposed formulation appears to be a post hoc way to connect the additive score to the genuinely FIM-based quantities presented in earlier sections.

**(W2) Dependence on a "calibration set":** Although the methods are described as "post-hoc," both the Tensor FIM Trace and Additive FIM Trace require access to a calibration set of ID data to fit the projection matrices $Q$ and $R$ (via PCA and/or LDA). This constitutes an additional training phase and requires at least partial access to ID data. Moreover, the performance of these methods is likely sensitive to the specific dataset, model, and selection of the calibration set. The paper, including Appendix A.2 ("Sensitivity to calibration set size"), does not clearly specify how this set is chosen or how its composition (not just its size) affects the results. If the selection is random, no error bars are provided to quantify the corresponding variability.

**(W3) Hyperparameters range and clarity:** The two subspace-based methods depend on several hyperparameters: the dimensions of $Q$ and $R$, the weights $\lambda_M$ and $\lambda_y$, and the algorithms used to construct $Q$ and $R$ (PCA or LDA). In the experiments, these choices are made via brute-force search, leaving unclear how they should be selected in practice or how robust the results are across different models and datasets. Overall, the discussion of hyperparameters seems to lack transparency. In addition to the ambiguity around the choice of the calibration set (see W2), Appendix A.3 ("Sensitivity to $\lambda$ parameters") is difficult to understand. It refers to the “top-left” and “bottom-right” of Figure 3 as regions of best performance, which does not correspond to the AUC scores actually displayed in the figure.


**(W4, minor) Novelty and related work:** I think the paper's claim that it introduces "a novel perspective on OOD detection through the lens of Riemannian information geometry" could be at least partially overstated. Previous works did use Fisher Information for OOD detection, one relatively well-cited example is Kwon et al. (2020) [1]. It would help if the authors clarified the difference from [1] and offer a general discussion of information-geometric methods for OOD detection.

Overall, my main concern is that the paper’s main proposed method does not fit coherently within the information-geometric framework developed earlier (see W1). This raises doubts about the usefulness of the framework itself. If the proposed methods demonstrated substantial empirical improvements over SOTA, this could be less concerning. However, since the paper’s primary goal appears to deepen understanding of the mechanisms behind OOD detection, this is a critical weakness.



## References

[1] Kwon, Gukyeong, et al. "Backpropagated gradient representations for anomaly detection." European conference on computer vision. Cham: Springer International Publishing, 2020.

**Questions:**

- Does the metric defined on the product manifold to justify the Additive FIM score retain any important properties of metrics defined above? What motivates calling it a “metric” rather than a heuristic combination of components?
- How is the calibration set selected (e.g., randomly sampled, curated subset)?
- Could the authors discuss the relationship to [1]?

---

> ### Author Response · Authors · 2025-11-20
> **Specific answer to Reviewer LmQv (part 1)**
>
> **Q1 and W1**:  The reviewer is right that the motivations for the additive score are somewhat subtle. We chose this score because we adopted the independence hypothesis between the uncertainty and the embedding signals. We refer the reviewer to our general response regarding the additive FIM-Trace, where we have provided a more detailed discussion on this point.
>
> **Q2 and W2**: We denote by “post-hoc” methods that do not update the parameters of a pre-trained model. In this sense, our approach remains strictly post-hoc, as all network weights $\theta$ are kept fixed, but the reviewer is right that some other definitions are possible. Moreover, relying on a subset of in-distribution data to compute test-time statistics is a standard practice in post-hoc baselines, including Mahalanobis [3] or KNN [4] approaches (which estimate class-conditional statistics) and ViM (which derives a principal feature subspace).
> In our work, the parameters $Q$,$R$ and related parameters are estimated on the training (ID) set without modifying the trained model, and to the best of our knowledge this complies with the requirements for a post-hoc method (as defined just above). Regarding the selection strategy, the calibration set is randomly sampled, and the following table reports results over three random seeds to assess variability, showing robustness to this random choice. The Uncertainty score has also been added for completeness.
>
>
>
> | **Calib. Size** | **Uncertainty** (AUROC ± Std Dev) | **Residual** (AUROC ± Std Dev) | **Proj. Norm** (AUROC ± Std Dev) | **Combined** (AUROC ± Std Dev) |
> | :---: | :---: | :---: | :---: | :---: |
> | 1,500 | 73.32 ± 0.65 | 37.78 ± 0.88 | 61.31 ± 0.82 | 76.04 ± 0.76 |
> | 2,500 | 76.38 ± 0.48 | 77.73 ± 0.76 | 62.43 ± 0.73 | 81.12 ± 0.65 |
> | 5,000 | 77.42 ± 0.34 | 79.06 ± 0.41 | 67.81 ± 0.53 | 82.00 ± 0.32 |
> | 10,000 | 78.45 ± 0.23 | 79.96 ± 0.25 | 71.74 ± 0.24 | 82.21 ± 0.26 |
> | 20,000 | 79.16 ± 0.09 | 80.19 ± 0.14 | 72.09 ± 0.13 | 82.99 ± 0.11 |
> | 40,000 | 79.29 ± 0.07 | 81.25 ± 0.10 | 72.14 ± 0.11 | 83.17 ± 0.08 |
> | 50,000 | 79.50 ± 0.05 | 81.46 ± 0.06 | 72.19 ± 0.08 | 83.22 ± 0.27 |

---

> > ### Author Response · Authors · 2025-11-20
> > **Specific answer to Reviewer LmQv (part 2)**
> >
> > **W3** : We thank the reviewer for their remark regarding the transparency of our hyperparameter selection and the inconsistencies in Appendix A.3. We acknowledge that our initial brute-force strategy lacked transparency.  We have revised this aspect by deriving an analytical solution tuning, as detailed in our General Response. Furthermore, we apologize for the error in the description of Appendix A.3; the reviewer correctly identified that the text contradicted the visual evidence in Figure 3 and we will correct this to align with the empirical results, as outlined in our General Response on Typo/Grid Search.
> >
> > **W4 and Q3**: We thank the reviewer for this reference. While our paper proposes an analysis on any given architecture, Kwon et al build their network around the proposed loss in a specific manner. In the related work, we will include  a synthesis of the following technical distinctions:
> >
> > 1. Both works operate on the insight that gradients w.r.t. weights contain rich information about the novelty of an input. Kwon et al. uses the *Fisher Kernel*, a function that measures the similarity between two data samples based on the gradients of a generative probability model.
> >
> > 2. Kwon et al. operate on Autoencoders and compute the gradient of the *reconstruction loss*: $g = \nabla_\theta \|x - \text{Dec}(\text{Enc}(x))\|^2$. This vector represents the sensitivity of the reconstruction error. In contrast, we target Probabilistic Classifiers and compute the gradient of the *log-likelihood*: $g = \nabla_\theta \log p(y|x)$. Formally, this is the Fisher Score.
> >
> > 3. Kwon et al. propose a **training-time** regularization ($\mathcal{L}_{grad}$) to constrain gradients to align with a stocked mean. They shape the manifold specifically and detect anomalies using the *cosine similarity* of gradients. Our paper assumes a given manifold which is the end result of a generic learning process and measures its intrinsic local sensitivity/curvature.
> >
> > ---
> >
> > This questioning encourages us to step back and articulate the broader vision that drives our work. We highlight the fact that the contribution is not just about a specific score, but about exploring a new idea in uncertainty quantification: the shift from **global** to **local** analysis. OoD detection, and more generally uncertainty quantification, has been dominated by global approaches: Bayesian/Ensemble methods [1],  estimating the posterior distribution and more specifically seeking for prediction's diversity.  This work helps establish a distinct but complementary paradigm. The core question shifts to: may information about uncertainty not only be captured by the diversity resulting from global exploration on the whole statistical manifold [2], but also encoded locally ? In this view, information geometry represents this idea. This local perspective is promising because it offers a path to capture uncertainty efficiently.
> >
> >
> > [1]  Balaji Lakshminarayanan, Alexander Pritzel, and Charles Blundell. “Simple and scalable predictive uncertainty estimation using deep ensembles”.
> > In: Advances in neural information processing systems 30 (2017).
> >
> > [2] Stanislav Fort, Huiyi Hu, and Balaji Lakshminarayanan. “Deep ensembles:
> > A loss landscape perspective”. In: arXiv preprint arXiv:1912.02757 (2019).
> >
> > [3] Kimin Lee, Kibok Lee, Honglak Lee, and Jinwoo Shin. A simple unified framework for detecting out-of-distribution samples and adversarial attacks. Advances in neural information processing systems, 31, 2018.
> >
> > [4] Yiyou Sun, Yifei Ming, Xiaojin Zhu, and Yixuan Li. Out-of-distribution detection with deep nearest neighbors. In International conference on machine learning, pp. 20827–20840. PMLR, 2022

---

> > > ### Comment · Reviewer_LmQv · 2025-11-26
> > >
> > > Thank you for the detailed rebuttal. While the authors’ response is generally clear and well-written, it unfortunately does not address my main concern regarding the motivation and information-geometric properties of the proposed additive score (W1).
> > >
> > > To be more specific why I see a problem here: a central motivation for this work, as detailed throughout Sections 3 and 4, is the information-geometric interpretation of the Fisher metric. In particular, one of the most important properties of the Fisher metric is its invariance under reparametrizations of the underlying statistical model. Although the multiplicative trace-based scores used in the paper are not fully invariant, simple reparametrizations such as $f_\theta \to c f_\theta$, $W \to \tfrac{1}{c} W$ only rescale the scores in a uniform way.
> > >
> > > However, the same reparametrization does not preserve the relative contributions of the components in the proposed additive metric. Under the same reparametrization, the uncertainty term $U_\theta$ remains unchanged, while both embedding-based terms are multiplied by $c^2$. Therefore, the decomposition and the final score can change significantly. To my mind, this sensitivity suggests that the additive construction departs too far from the rationale behind using the Fisher–Rao geometry in the first place.
> > >
> > > Given that another reviewer has also noticed this issue, I believe it is an important conceptual point. While I see that the method is justified by its empirical performance, I am still not convinced that information geometry is the right framework to explain the success of such a method. Therefore, I am currently inclined to maintain my score, though I remain open to further discussion with the reviewers.

---

> > > > ### Author Response · Authors · 2025-11-27
> > > > **Response for the impact of rescaling**
> > > >
> > > > We thank the reviewer for engaging in discussion and for this theoretical observation. You are correct that for an arbitrary fixed set of weights $(\lambda_y, \lambda_M)$, the additive score would lose the invariance property under rescaling. However, the critical detail that prevents this scenario is that a rescaling operation of the network  requires a correct evaluation of $(\lambda_y, \lambda_M)$ following the analytical formulas we presented in the general response above (which replace the naive grid search).
> > > >
> > > > Let us demonstrate below that the updated values $(\lambda’_y, \lambda’_M)$ following rescaling keep the invariance mentioned by the reviewer. Recall that:
> > > >
> > > > $$
> > > > \lambda_y = \sqrt{\frac{\text{Var}(U)}{\text{Var}(y)}}, \quad
> > > > \lambda_M = \sqrt{\frac{\text{Var}(U)}{\text{Var}(M)}}
> > > > $$
> > > > up to a sign. Consider the rescaling reparameterization mentioned by the reviewer: $f_\theta' = c f_\theta$ (where $c \in \mathbb{R}^*$), assuming the classifier weights adapt ($W' = W/c$) to maintain invariant predictions.
> > > >
> > > > 1 The uncertainty $U$ (based on probabilities) remains invariant: $U' = U$.
> > > >
> > > > 2 The embedding-based terms (squared norms) scale quadratically: $M' = \|c f_\theta\|^2 = c^2 M$ and $y' = c^2 y$.
> > > >
> > > > 3 Consequently, their variances scale by $c^4$: $\text{Var}(M') = c^4 \text{Var}(M)$.
> > > >
> > > > If the $\lambda_M$ hyperparameter value were computed on the rescaled data, it would adapt as follows:
> > > >
> > > > $$
> > > > \lambda'_M = \sqrt{\frac{\text{Var}(U')}{\text{Var}(M')}} = \sqrt{\frac{\text{Var}(U)}{c^4 \text{Var}(M)}} = \frac{1}{c^2} \lambda_M
> > > > $$
> > > >
> > > > The scaling factor $c^2$ from the feature magnitude is precisely cancelled by the $1/c^2$ scaling of the hyperparameter:
> > > >
> > > > $$
> > > > \lambda'_M M'(x) = \left(\frac{1}{c^2} \lambda_M\right) (c^2 M(x)) = \lambda_M M(x)
> > > > $$
> > > >
> > > > The same applies to the residual term.Thus, the resulting additive score is invariant under rescaling, ensuring that the additive decomposition remains consistent. We hope that this clarification fully addresses the reviewer’s concern. The invariance property will be explicitly stated in the final version to ensure that similar questions from future readers are also resolved.

---

### Official Review · Reviewer_BE7f · 2025-10-26

**Soundness:** 3
**Presentation:** 3
**Contribution:** 3
**Rating:** 8
**Confidence:** 3

**Summary:**

This paper reframes Out-of-Distribution (OoD) detection using Riemannian information geometry, hypothesizing that OoD inputs exhibit higher "local Fisher-Rao sensitivity" than in-distribution data. The authors first provide a novel theoretical proof (Theorem 5.1) that this geometric sensitivity, quantified by the trace of the Fisher Information Matrix (FIM), mathematically decomposes into the product of two common OoD signals: feature magnitude and output uncertainty. Finally, after analyzing this product's limitations, the paper proposes a "product manifold" construction to provide a formal geometric justification for robust additive scores, which motivates their final, competitive "Additive FIM Trace" method.

**Strengths:**

* The paper's primary strength is providing a new, unifying geometric language for OoD detection and offers what appears to be the first formal geometric justification for the empirical success of robust additive OoD detectors (like the SOTA method ViM)

* The paper introduces a critical refinement of existing feature-space methods. This is a novel and significant finding that validates a key design choice in its final, competitive "Additive FIM Trace" score

* The experimental validation is comprehensive and thorough

**Weaknesses:**

* The clarity of section 5.3.1 could be improved (I found it difficult to follow and verify). In particular the introduction of the product manifold $M = U \times U \times U$ seems unmotivated. Can you add a high-level description of this section?

* The role of the last layer, while clear from a practical standpoint, seems theoretically somewhat arbitrary. Can the authors briefly comment on the role of the remaining parameters (even if the formula become now intractable)? Could they theoretically be added to the statistical manifold? What additional information would this provide?

* The hyper-parameters $\lambda_M$ and $\lambda_y$ need to be tuned. Can the authors comment on the range of these parameters for different settings? Are there "good default" or the tuning results in widely different values for different settings?

**Questions:**

See weakness section.

---

> ### Author Response · Authors · 2025-11-20
> **Specific answer to Reviewer BE7f**
>
> **W1**: ​​We thank the reviewer for this feedback. We agree that the motivation for the product manifold was dense. We will add the following high-level intuition in the revised version:
>
> 1. Just as statistically independent variables are modeled by a joint distribution that factorizes, geometrically independent (orthogonal) spaces are modeled by a **Product Manifold** $M := \mathcal{U} \times \mathcal{U} \times \mathcal{U}$. The tangent space of a product manifold is the direct sum of the component tangent spaces ($TM \cong T\mathcal{U} \oplus T\mathcal{U} \oplus T\mathcal{U}$), which implies that the components are geometrically decoupled and no cross-term interactions occur.
>
> 2.  On this product structure, we need to define a specific metric on each copy of
> $\mathcal{U}$ to isolate the contribution of each signal while preserving the underlying geometry. We achieve this via conformal transformations of the original Fisher metric. By rescaling the metric on each copy of $\mathcal{U}$, we define a new local geometry where the trace corresponds exactly to our signal of interest ($U, y$, or $M$). We can even extend this to pseudo-conformal transformations, by allowing negative factors. This results in a **Pseudo-Riemannian**[1] metric on the product space where sensitivity/curvature is still well-defined.
>
> **W2**: We thank the reviewer for their question. We provide a detailed discussion on the Last-Layer Restriction in our General Response. In essence, while extending the manifold to include all parameters is theoretically defined, it would render the exact trace computation intractable. Furthermore, empirical evidence from related work [2] suggests that the most discriminative sensitivity signal for OOD detection is surprisingly concentrated in the final linear projection.
>
> **W3**: We thank the reviewer for their question. Regarding the tuning ranges and good defaults for $\lambda_M$ and $\lambda_y$, we have detailed this question in our General Response on hyperparameter tuning where we explain that we will replace the numerical grid search objective by an analytical solution derived from variance normalization.
>
> [1] Sylvestre Gallot, Dominique Hulin, Jacques Lafontaine, et al. Riemannian geometry. Vol. 2. Springer, 1990.
>
> [2] Rui Huang, Andrew Geng, and Yixuan Li. “On the importance of gradients for detecting distributional shifts in the wild”. In: Advances in Neural Information Processing Systems 34 (2021), pp. 677–689.

---

> > ### Comment · Reviewer_BE7f · 2025-11-26
> >
> > Thank you for your explanations. As a result, I am maintaining my score as a good paper to be accepted.

---

### Official Review · Reviewer_ba3P · 2025-11-01

**Soundness:** 3
**Presentation:** 3
**Contribution:** 3
**Rating:** 8
**Confidence:** 4

**Summary:**

This work proposes an OOD detection strategy that makes use of information geometric guidance on the last layer of deep neural network model for classification tasks. The strategy implements itself into three different proxies, all based on using trace of Fisher Information Matrix (FIM) as their foundations. The three proxies differ by how the last layer parameter space is decomposed, such that different hyperparameter weights for the traces from each decomposed portions can be applied. The weights, in addition to a couple of projection matrices to decompose the FIM are significant hyperparameters of the proposed OOD detection method. Empirical validations are comprised of ImageNet-based and CIFAR-based scenarios, in which the proposed method is compared to nine other OOD detection methods.

**Strengths:**

The work in this manuscript proposes a mathematically grounded strategy to approach OOD detection problem for classification tasks. Albeit limited to last layers only, the actual implementation comes in three flavors, making use of trace(FIM) directly, and then decomposing FIM into two pieces, then into three pieces. The last decompostion, which explicitly takes residual orthogonal subspaces seems to make the strategy wholesome and beautifully closed. This gradual improvement makes this work contain educational value as well as researchwise novelty.

The mathematical complications under the hood of the proposed algorithm are nicely placed in the appendices, in addition to easy-to-read comparison to IGEOOD which also makes use of local geometry of models wrt their parameters. I also appreciate significant portion of work on sensitivity analyses in the appendix.

Empirical validation section is much richer than a typical theory-oriented manuscript. This is a desirable stance, as the less-mathematically-equipped audience and those who values practical impact much more will find the work more acceptable.

**Weaknesses:**

The key idea would be much more intuitively understood if theorem 5.1 would accompany the contextualized connection to the strategy found in Eq. 8. I found relevant portions in Appendix D, which was a delight, but at the same time, a dismay that this has to be placed in the appendix rather than places like lines 264-266. Also, the opening of the paragraph on lines 328-332 is personally enjoyed but may throw some readers away due to its mathematical brevity. There are some assumptions simply given in that paragraph without any contextual explanation (e.g. assuming nonzero fraction in line 332), whose meaning may be justified intuitively and concisely.

It seems that this method relies heavily on a good choice of D_val dataset to calibrate the key parameters \lambda's, R, and Q. The sensitivity to the size of the D_val is in the appendix, but as the method claims to be an OOD detection method that does not require OOD dataset, I think more thorough analysis (either theoretical or empirical) would provide readers much more insight to this method and OOD detection in general.

Please correct. Two minor typos I found:
1) on lines 302-303. "in Eq. equation 9".
2) in the equation for F''' in lines 334-335. L2norm-squared looks weird after the first f_{\hat{\theta}}(x) (don't you want it embrace the entire f(x) - RR^T f(x), which is already there?)

**Questions:**

The robustness of the additive decomposition of geometric space from last layer may depend on a well-chosen set of relative weights \lambda's. The sensitivity analysis results in Figure 3 seems to agree on this, and at the same time suggesting some narrow goldilocks zone, which unfortunately seems to be per-dataset, per-model, a posteriori information which requires grid search. Would there be any alternatives that make this hyperparameter search more theory-guided (more post-hoc, if I were to hijack the term)?

This is mentioned in the limitation. But let me raise it once more. OOD behavior may be also due to factors hidden in the learnable feature (i.e. the non-last-layer of the NN models). How would this information geometric idea extend to the deeper layers?

---

> ### Author Response · Authors · 2025-11-20
> **Specific answer to Reviewer ba3P**
>
> We appreciate the reviewer’s positive feedback regarding
> Appendix D.  The full proof was placed in the appendix to avoid overloading the main text
> with technical details that may not be necessary for all readers.
> However, we will include a concise proof sketch in the revised version, placed immediately after Theorem 5.1, providing a clear description of the
> core ideas and structure of the argument. For the assumption given in the paragraph 328-332, we will add a more pedagogical explanation in the future revised version.
>
> Regarding the calibration of key parameters. We agree that understanding the dependence on $D_{val}$ is crucial. Regarding the calibration of the $\lambda$'s, we refer the reviewer to our General Response on Hyperparameter Tuning. We have addressed this concern with a principled analytical solution.
>
> Concerning the typos, we thank the reviewer for their reading and for identifying these specific errors.  We will include these corrections and will ensure that the redundancy and the notation in the $F'''$ equation are fixed in the revised manuscript.
>
>
>
> **Q1**: Thank you for raising this point. We agree that relying on a posteriori grid search was a limitation. As noted in our general response on hyperparameter tuning, we now replace it with an analytical calibration that removes the need for grid search and follows the theory.
>
> **Q2**: We thank the reviewer for their question. Our analysis focus on the block of the Fisher matrix corresponding to the last layer because this is the only case where we can derive a closed form expression (Theorem 5.1) and an explicit decomposition with a feature magnitude term and a predictive uncertainty term. However, except the significant computational cost,  nothing stops us from defining the Fisher matrix in a more general case. Concretely for any subset of parameters $\theta_\ell$ we have the expression:
> $$
> F_{\theta_\ell,x}(\hat \theta)=  \mathbb{E} \big[\nabla_{\theta_\ell}\log p(y\mid x,\hat \theta)\,
> \nabla_{\theta_\ell}\log p(y\mid x,\hat \theta)^{\top}\big],
> $$
> where $y\sim p(\cdot\mid x,\hat \theta) $. Its trace $\mathrm{Tr}(F_{\theta_\ell,x}(\hat \theta))$ measures the Fisher-Rao sensitivity with respect to the layers selected, exactly as we do for $F_{W,x}(\hat \theta)$  in equation 5. In the general answer, we provide two strong arguments already present in the literature which indicate that going beyond the last layer is not helpful.

---

### Official Review · Reviewer_6tV2 · 2025-11-03

**Soundness:** 4
**Presentation:** 3
**Contribution:** 2
**Rating:** 6
**Confidence:** 3

**Summary:**

This paper is presenting a new Out-of-Distribution (OoD) detection method (post-hoc method) grounded in information geometry (Riemannian manifold). The authors model a network's predictive distribution as a statistical manifold and hypothesize that OoD inputs exhibit higher "local Fisher-Rao sensitivity" than in-distribution (ID) inputs. The paper's first main contribution is Theorem 5.1, which provides a closed-form expression for this sensitivity obtained from the last layer's statistics. Later authors criticize the multiplicative nature of the score and moving into additive version. Competitive with SOTA methods is shown.

**Strengths:**

- Theorem 5.1 provides an analytical, and geometrically-grounded reason why feature magnitude and output uncertainty are such effective signals for OoD detection. It successfully bridges the gap between abstract geometric theory and common empirical practice.
- The paper clearly explains why a simple product of (magnitude * uncertainty) is a flawed score: ID inputs are (High * Low) while OoD inputs are (Low * High), and these products can be indistinguishable .
- ablation studies are rich.

**Weaknesses:**

- true motivation is empirical rather than quantitative/theoretical.
- does U(x) provide any benefit?
- Wang et al 2022 paper has extremely similar formulation on additive score that authors are presenting. Clarification needed. Conceptual overlap is very strong. Isolate contribution from this paper.
- why PCA is chosen for class subspace Q? how about other methods like unsupervised methods ?

**Questions:**

weaknesses section have self-contained questions for each item.

---

> ### Author Response · Authors · 2025-11-20
> **Specific answer to Reviewer 6tV2**
>
> **W1**: We tried to synthesize in the General answer/Interpretation of the Additive score why the score we propose is methodologically founded in terms of local sensitivity  analysis. In the response for Reviewer LmQv, we also explain how we perceive the relevance of local sensitivity analysis compared to the more popular global loss landscape exploration strategies. Of course, this implies that no single strategy is ``better’’ than others, and that the main concern should be to be as transparent as possible about the hypotheses we adopt in a given framework.
>
> **W2**:   Yes, the uncertainty term $U(x)$ provides a distinct and complementary benefit for OoD detection, compared to the feature-based terms ($M(x)$ and $y(x)$). While the feature-space components $M(x)$ and $y(x)$ measure -distance from the ID manifold, $U(x)$ measures *ambiguity*. $U(x)$ captures this decision-level uncertainty that feature information misses [1].
>
> **W3** : Although there are indeed similarities with ViM, which we use as a strong baseline, the main distinction is the following.
>
> ViM constructs a virtual logit from the residual and requires a heuristic scaling factor $\alpha$ to manually match the magnitude of the original logits. In contrast, the components ($U, M$) are derived analytically from our geometrical analysis. We proved that, when considering the learning process in terms of its instability on a statistical manifold, local sensitivity naturally decomposes into these specific signals. The score follows directly from the analytical expression of the FIM trace and the orthogonality assumption, rather than from any heuristic choice.
>
>
> **W4**: We chose Principal Component Analysis (PCA) over manifold learning methods (like t-SNE or UMAP) for the following reasons.
>
> PCA is the natural linear algebraic tool designed to analyze such covariance structures: it diagonalizes the covariance matrix to identify the orthogonal directions of maximum variance. Therefore, applying PCA to constructing the subspace $Q$ is mathematically consistent: it gives the principal axes of the Fisher information Matrix (the directions of highest sensitivity) in a linear basis. Using Using non-linear neighbor-embedding methods like t-SNE or UMAP introduce additional complexity which needs to be justified. They rely on sensitive hyperparameters (e.g., perplexity, number of neighbors, min dist). By not requiring the user to tune more hyperparameters for the subspace construction, this ensures that our method remains simple to use.
>
>
> [1] Jarrod Haas, William Yolland, and Bernhard Rabus. “Exploring simple,
> high quality out-of-distribution detection with l2 normalization”. In: arXiv
> preprint arXiv:2306.04072 (2023)

---

> > ### Comment · Reviewer_6tV2 · 2025-11-24
> > **I am satisfied with the answers**
> >
> > I thank the authors for detailed responses to me and also some other responses by other reviewers. I was already positive about this paper and willing to increase my score.

---

### Official Review · Reviewer_1qwa · 2025-11-09

**Soundness:** 2
**Presentation:** 2
**Contribution:** 3
**Rating:** 4
**Confidence:** 2

**Summary:**

The paper proposes a post-hoc OoD detection method based on the per-input trace of the Fisher Information Matrix (FIM) restricted to the final-layer network weights. This acts as a proxy for model sensitivity, which the authors claim is greater for OoD versus ID signals. For a linear softmax head, the authors show that this “standard FIM trace” factorizes into the product of the L2-norm of the final layer features, and a predictive uncertainty term, unifying two well-known OoD signals. They further introduce a more performant “tensor-factorized” variant of this score that involves projecting the final-layer FIM onto a learned subspace, which is constructed from feature and prediction statistics. Finally, they present a more robust additive version of the tensor-factorized score, which they demonstrate is competitive with current SOTA post-hoc OoD detection methods on CIFAR and ImageNet-style benchmarks.

**Strengths:**

The paper gives an interesting information-geometric justification for the use of two popular OoD signals (feature magnitude, prediction uncertainty). The tensor-factorized subspace approach appears novel and meaningful, and the underlying "orthogonality hypothesis" (if further supported) could provide a useful principle for designing new OoD detection methods. The empirical evaluation is thorough, and demonstrates that the final additive version of their OoD score consistently attains performance which is competitive with similar state of the art methods. The ablation studies are also reasonably extensive, although some of the details could benefit from additional clarification (see questions).

**Weaknesses:**

1. The geometric interpretation of the additive score (section 5.3) reads somewhat like a post-hoc justification. That is, the product manifold and metric seem to be engineered so that the trace reproduces the score, rather than the design of the score being informed by some underlying geometric principle. It would be helpful to clarify whether this product manifold is in some sense “canonical”, and whether it offers any insights/uses beyond re-expressing the additive score in geometric notation.

2. The orthogonality hypothesis (Appendix B) proposes that the tangent subspace corresponding to the final-layer weights can be decomposed into two approximately orthogonal subspaces: one determined by the layer inputs (features) and one by the outputs (class predictions). This is an intriguing and potentially quite insightful claim; however, it does not appear to be supported by much empirical or theoretical evidence. Some further investigation in this direction could strengthen the paper.

3. The paper would benefit from a careful editing pass. There are multiple small typos, some notation issues, and also some potential errors in the ablation studies. Several of these have been highlighted in the questions section.

**Questions:**

Ablation Studies:

1. Impact of subspace construction (A.1): it is mentioned that a partial least squares approach is examined for computing the feature subspace. However, this is not included in the results (Table 3). Furthermore, the baseline (no projection) yields slightly different AUROC contributions for the residual and projected norms, which does not make sense. Have these two been mixed up?

2. Sensitivity to score weights (A.3): it is mentioned that the best AUROC performance is attained when the projected norm is high and residual is low (or vice versa), however the graph shows that this occurs along the diagonal (i.e., when both residual and projected norm are high).

Miscellaneous:

1. The use of notation of the form $X_{\text{Long Text}}$ is a little awkward. For example, something like $S_{x_{OoD}}$ and $S_{x_{ID}}$ could be replaced by $S_x^O$ and $S_x^I$, while $G_{probability}$ and $G_{feature}$ could be $G_p$ and $G_f$.

2. In lines 912-914, the terms $p_{k,\hat{\theta}}(x)(x)$ seem like they should be $p_{k,\hat{\theta}}(x)$.

---

> ### Author Response · Authors · 2025-11-20
> **Specific answer to Reviewer 1qwa**
>
> **W1** : We refer the reviewer to our general response regarding the additive FIM-Trace, where we have provided a detailed discussion on this point
>
> **W2** : We thank the reviewer for this comment. As detailed in the general response, we generalize our hyperparameter tuning beyond the grid-search,  and we show that the independence/orthogonal assumption is key in order to link the loss to the scale-balancing objective of the variance of each signals $U, M$ ands $y$. We will add this discussion in the final version.
>
> **W3** : We thank the reviewer for their careful reading and for highlighting these issues. We apologize for the typos and inconsistencies and will proofread the manuscript to correct them in the revised version. For the ablation error, we refer the reviewer to the general answer.
>
> **Q1** :  The reviewer is correct: the Partial Least Squares (PLS) variant was initially implemented but was removed from the final set of experiments due to instability and a lack of performance. The text was not updated accordingly, leading to some inconsistencies noted in Table 3. We will revise Appendix A.1 accordingly. Additionally, regarding the Identity baseline results, the reviewer is right to point out the inconsistency. Mathematically, with no projection ($R=I$), the residual vector is null, implying a Residual AUROC of exactly $50.00\%$. The reported value ($57.00$) was a transcription error. In contrast, the Projected Norm correctly reflects the full feature magnitude ($\|f(x)\|_2^2$), so its score ($61.07$) is valid. We checked the remaining reported values, to make sure that no other such typos remain. We will correct the Residual value to $50.00\%$ in the revised table.
>
> **Q2** : We apologize for the error in the description of Appendix A.3; the reviewer correctly identified that the text contradicted the visual evidence in Figure 3 and we will correct this to align with the empirical results, as outlined in our General Response on Typo/Grid Search.
>
> **Misc** : noted, thank you for your vigilance.

---

### Author Response · Authors · 2025-11-20
**Response to shared reviewer remarks (part1)**

Dear reviewers, we thank you for your constructive feedback and effort. We start our response by grouping and answering here the shared remarks in order to minimize redundancy and keep the discussion clear.

# Hyperparameter tuning

A shared concern among the reviewers relates to the grid-search we employed for tuning the $\lambda$ hyperparameters. This default brute force approach was justified by the non-convex objective, but following the reviewer remarks we realized that an embarrassingly obvious analytical solution exists, and we will update the manuscript accordingly. We detail this update below.

## Analytical hyperparameter search

Our ID-only tuning objective is to balance the scales of the independent components ($U, y, M$) so they contribute equitably to the final score. The grid-search approach was a simple method to minimize the following objective on the ID validation set:
$$
\mathcal{L}(\lambda_y, \lambda_M) = \left( \text{Var}[\lambda_y \cdot y(x)] - \text{Var}[U(x)] \right)^2 + \left( \text{Var}[\lambda_M \cdot M(x)] - \text{Var}[U(x)] \right)^2
$$

Since $\text{Var}[\lambda_y y] = \lambda_y^2 \text{Var}[y]$, this objective $\mathcal{L}$ is a 4th-degree polynomial in $\lambda_y$ and $\lambda_M$. This immediately yields the analytical solution:

$$
|\lambda_y| = \sqrt{ Var_{ID}(U) / Var_{ID}(y) }  \quad \text{and} \quad |\lambda_M| = \sqrt{ Var_{ID}(U) / Var_{ID}(M) }
$$

We chose the signs based on the signal contribution analysis detailed in Appendix D.

One might wonder whether minimizing $\mathcal{L}$ is an interesting way to balance the score. The choice of minimizing $\mathcal{L}$ is supported by the interpretation that under the independence assumption, one can effectively balance the variance scale of all the signals. This follows from the assertion that in the absence of knowledge or any further assumption about OoD, this is the natural choice.  The developed expression for the variance of the additive score $S$ is:

\begin{align*}
    \text{Var}(S) = & \quad \text{Var}(U) + \lambda_y^2 \text{Var}(y) + \lambda_M^2 \text{Var}(M) + 2\lambda_y \text{Cov}(U, y) + 2\lambda_M \text{Cov}(U, M) + 2\lambda_y\lambda_M \text{Cov}(y, M)
\end{align*}

Under the orthogonality assumption (Eq.(10) \& App.~B), the covariance terms vanish. Then, the global variance boils down to:
$$
\text{Var}(S) = \text{Var}(U) + \lambda_y^2 \text{Var}(y) + \lambda_M^2 \text{Var}(M)
$$
To ensure that each component contributes equally to this global variance, one must satisfy:

1. $\lambda_y^2 \text{Var}(y) = \text{Var}(U)$
2. $\lambda_M^2 \text{Var}(M) = \text{Var}(U)$

**Conclusion:** This leads to the exact same analytical solution as minimizing $\mathcal{L}$. Thus, the orthogonality assumption guarantees that minimizing the simple marginal error $\mathcal{L}$ effectively balances the total variance of the score.

Tables 1, 2 and 3 compare the performance of the proposed Additive FIM Trace using the initial Grid Search (GS) strategy using the analytical solution (keeping the same seed). The analytical solution method yields performance matching in practice the naive grid search.


Comparison of Tuning Methods on ImageNet (ViT-B/16)
| **VITB16** | **ImageNet-O** (AUROC / TNR95) | **iNaturalist** (AUROC / TNR95) | **Places365** (AUROC / TNR95) | **SUN** (AUROC / TNR95) |
| :--- | :---: | :---: | :---: | :---: |
| Additive FIM Trace (Grid Search) | 92.74 / 68.94 | 99.03 / 98.67 | 89.97 / 60.81 | 93.66 / 68.81 |
| **Additive FIM Trace (Analytical)** | 92.61 / 68.25 | 98.82 / 98.41 | 89.51 / 60.17 | 93.38 / 67.98|

---
---

Comparison of Tuning Methods on ImageNet (ResNet50)
| **Resnet50** | **ImageNet-O** (AUROC / TNR95) | **iNaturalist** (AUROC / TNR95) | **Places365** (AUROC / TNR95) | **SUN** (AUROC / TNR95) |
| :--- | :---: | :---: | :---: | :---: |
| Additive FIM Trace (Grid Search) | 71.97 / 22.38 | 90.78 / 54.84 | 83.21 / 40.56 | 85.89 / 40.08 |
| **Additive FIM Trace (Analytical)** | 71.77 / 22.41 | 90.61 / 54.69 | 82.93 / 40.33 | 85.76 / 39.95 |

---
---

Comparison of Tuning Methods on CIFAR/SVHN (ResNet18)
| **Resnet18** | **ID: CIFAR-10 vs. SVHN** | **ID: CIFAR-10 vs. C-100** | **ID: CIFAR-100 vs. C-10** | **ID: CIFAR-100 vs. SVHN** |
| :--- | :---: | :---: | :---: | :---: |
| Additive FIM Trace (Grid Search) | 91.25 / 68.99 | 87.17 / 54.36 | 74.84 / 15.42 | 81.05 / 35.78 |
| **Additive FIM Trace (Analytical)** | 91.28 / 68.88 | 87.23 / 54.47 | 74.38 / 15.03 | 80.91 / 35.35 |

---

> ### Author Response · Authors · 2025-11-20
> **Response to shared reviewer remarks (part2)**
>
> # Interpretation of the Additive score
>
> We agree with the reviewers that, as pointed out in the Limitations section, the additive score is not canonical as there exists an infinity of possible conformal/pseudo-conformal metric transformations. We chose this construction because we adopt the hypothesis of independence between the uncertainty and embedding signals. This score is naturally mathematically modeled by the product manifold.
>
> On this product manifold, equipped with a tensor-based pseudo-Riemannian/Riemannian metric, the notion of local sensitivity (defined as the trace of the metric) is well-defined. The statistical interpretation is based on the independence of the metrics on each component. Since this additive score is less intuitive than the base FIM setup, we will give a more detailed overview of the statistical properties of this construction in the revised version. Note also that, beyond mathematical formulation, the additive score is supported by empirical validation: it was able to achieve better results than the Tensor FIM Trace.
>
> # Typo in Grid Search Ablation
>
> We thank the reviewers for carefully identifying the paper inconsistencies and typos in general.  In particular, we acknowledge the mismatch between the textual description and the plotted  results. The text incorrectly stated that optimal AUROC was achieved when one component was high and the other low. As the reviewers correctly note, the  heatmap shows the opposite: the diagonal region (high projected norm and  high residual) yields the best performance. We will update the corresponding  paragraph in Appendix~A.3 to match the empirical observation.
>
> # Discussion on Last Layer restriction
>
> We thank the reviewers for their questions about the potential benefit of considering features in deeper layers. We chose to restrict our analysis to the final layer for the two following reasons:
>
> 1. As noted by the reviewers, gradient-based methods have explored this dimension. In the works of **GradNorm** [1], the authors compared the efficacy of gradients extracted at various depths (see their **Table 2**). Their conclusion is: using parameters from the **last linear layer only** outperforms not only intermediate layers but also the aggregation of all network parameters ("All params"). This suggests that the most discriminative signal for OOD detection is concentrated in the final layer.
>
> 2. One of our contributions is to provide an exact analytical expression for sensitivity. This is made possible by the linear structure of the final layer. Extending this analysis to deeper layers would make the calculation of the FIM trace intractable. For $N$ parameters, it would require to build or to approximate the trace of an $N^2$ size matrix, for each input, which is clearly intractable and depending on the backbone architecture.
>
>
>
> [1] Rui Huang, Andrew Geng, and Yixuan Li. “On the importance of gradients for detecting distributional shifts in the wild”. In: Advances in Neural Information Processing Systems 34 (2021), pp. 677–689

---

### Author Response · Authors · 2025-12-03
**Submission of Revised Manuscript**

Dear Area Chair and Reviewers,

We have uploaded a revised version of our manuscript that incorporates the valuable feedback and suggestions raised during the discussion period. We are grateful for the opportunity to improve our work and summarize below the specific sections where significant modifications have been made:

1. **Analytical Hyperparameter Tuning & PCA design (Section 5.3.1)**: We have replaced the initial grid-search approach with a rigorous analytical solution derived from the variance balancing of components (Propositions 5.3 & 5.4 and their associated proof in Appendix E).
Accordingly, we have updated the reported scores to reflect this generalization (Section 6).
Importantly, Proposition 5.4 establishes the score's invariance under reparameterization, directly addressing the theoretical concern raised by Reviewer **LmQv**.
This addresses the main concern regarding hyperparameter transparency raised by **multiple reviewers.**
We also clarified the use of PCA for matrix construction, a point discussed with Reviewer **6tV2**.

2. **Extended Results (Appendix A)**: We have added detailed tables in the Appendix, including the Uncertainty component for completeness, to provide a transparent breakdown of the method's performance and robustness to random choice of validation set, as requested by Reviewer **LmQv**. We also added in Appendix D a toy experiment to illustrate the intrinsic geometric dynamics of Fisher-Rao sensitivity and to explicitly visualize the magnitude-uncertainty trade-off. By exposing how low uncertainty suppresses the total score in standard multiplicative metrics. This example addresses the remarks of reviewers **LmQv**   and **1qwa** about the necessity of the proposed additive decomposition to decouple these conflicting signals for effective Out-of-Distribution detection.

3. **Orthogonality Hypothesis (Appendix B and F)**: We have rewritten this section to clarify and generalize the intuition and formalism behind the product manifold construction, local curvature and orthogonal hypothesis. We also added formal definitions for the direct sum of tangent spaces, as requested by Reviewer **BE7f and others**.

4. **Proof Sketch (Section 5.2)**: A sketch of the proof for Theorem 5.1 has been added immediately after the theorem to improve intuition, as suggested by Reviewer **ba3P**.

5. **Ablation Study Corrections (Appendix A)**: We corrected the description in Appendix A.3 to match the empirical results shown in Figure 3, and corrected typos in Table 3 (Appendix A.1), addressing points raised by Reviewers **1qwa** and **LmQv**.

6. **Comparisons & Related Work**: We have added a more comprehensive comparison between our method and ViM (Appendix C), and mentioned Kwon et al. (2020) (Section 2) to clarify the positioning of our work relative to prior Fisher-based approaches, as requested by Reviewers **6tV2** and **LmQv**.

7. **Global vs. Local Perspective (Conclusion)**: As discussed with Reviewer **LmQv**, we expanded the concluding remarks to introduce the broader paradigm shift suggested by our work: moving from global uncertainty quantification (e.g., Deep Ensembles) to local geometric sensitivity analysis.

8. **General Corrections**: We have proofread the manuscript to correct the typos and minor inconsistencies, particularly those pointed out by Reviewers **1qwa** and **ba3P**.

We express our gratitude to the Area Chair and all reviewers for their engagement with our paper, which has been significant in strengthening the quality and clarity of our manuscript.

Best regards,

The Authors.

---

### Meta-Review · Area_Chair_rT49 · 2026-01-08

**Summary:**

This paper received 5 reviews that were quite mixed but leaning positive (8, 8, 6, 4, 4) .  Several reviewers found the theoretical presentation in the paper to be quite compelling, and they found the empirical results and ablations strong and convincing.  A major concern among the dissenting reviewers was over mathematical rigor of the work and in particular highlighted that the method that does end up working well isn't really justified by the theory.   The theory points to a multiplicative score, the Fisher information trace, but the authors found that it didn't work quite as well as an additive approach, which is more heuristic and isn't really justified by the theory presented.  Overall, it seems like after the discussion period there would be at least three scores of 8, one of 4 and one is a bit unclear.  I'm sympathetic to the concerns of the holdout reviewer, who is skeptical of the leap from the justified multiplicative score the less-justified additive one.  I also am somewhat concerned about novelty.  There is a significant literature on approximate-Bayesian inference using approximations to the Fisher-information (or local curvature), including methods such as KFAC, or Laplace and diagonal Laplace.  There are also natural gradient based OOD methods.  However, there is a consensus of reviewers who believe the paper should be accepted, and they argue that particular novel contributions are both the analytical score and the theoretical analysis (which is interesting even if it doesn't end up being exactly useful).   That seems to tip the paper over the accept bar in my opinion.

**Reviewer Concerns:**

It seems clear that reviewer lmqv wasn't convinced enough by the rebuttal to change their score: "Thank you for the detailed rebuttal. While the authors’ response is generally clear and well-written, it unfortunately does not address my main concern regarding the motivation and information-geometric properties of the proposed additive score (W1)."

Reviewer 6tV2 seemed to be positive on changing their score ("I thank the authors for detailed responses to me and also some other responses by other reviewers. I was already positive about this paper and willing to increase my score.").  It's plausible this would be raised to a 8, although I  don't think this review itself was particularly substantive.

**Reviewer Scores:**

Clearly lmqv would keep their score at 4 and 6tV2 would increase their score.  It seems hard to forecast 1qwa, but it's plausible they might increase their score by 1 (or keep it fixed).

---

### Decision · Program_Chairs · 2026-01-26

Accept (Poster)